# Atypical rhizobia trigger nodulation and pathogenesis on the same legume hosts

Kévin Magne[1,2,9], Sophie Massot[1,2], Tifaine Folletti[3], Laurent Sauviac[3], Elhosseyn Ait-Salem[1,2], Ilona Pires[1,2], Maged M. Saad [4], Abdul Aziz Eida [4], Salim Bougouffa [5], Adrien Jugan[1,2], Eleonora Rolli[1,2,10], Raphaël Forquet[6], Virginie Puech-Pages[7,8], Fabienne Maillet[3], Gautier Bernal [1,2], Chrystel Gibelin[3], Heribert Hirt [4], Véronique Gruber[1,2], Rémi Peyraud [6], Fabienne Vailleau[3], Benjamin Gourion [3] ✉ & Pascal Ratet [1,2] ✉

The emergence of commensalism and mutualism often derives from ancestral parasitism. However, in the case of rhizobium-legume interactions, bacterial strains displaying both pathogenic and nodulation features on a single host have not been described yet. Here, we isolated such a bacterium from *Medicago* nodules. On the same plant genotypes, the T4 strain can induce ineffective nodules in a highly competitive way and behave as a harsh parasite triggering plant death. The T4 strain presents this dual ability on multiple legume species of the Inverted Repeat-Lacking Clade, the output of the interaction relying on the developmental stage of the plant. Genomic and phenotypic clustering analysis show that T4 belongs to the nonsymbiotic *Ensifer adhaerens* group and clusters together with T173, another strain harboring this dual ability. In this work, we identify a bacterial clade that includes rhizobial strains displaying both pathogenic and nodulating abilities on a single legume host.

Microbial symbionts are currently thought to emerge from transitions along the parasite-mutualist *continuum* and these shifts are mainly driven by genetic, environmental and ecological changes[1]. Shifts from mutualism to parasitism appear rare while the emergence of commensal and mutualistic organisms from parasitic ancestors are more frequent[1,2]. In the case of the rhizobium-legume mutualism, rhizobia, which trigger nitrogen-fixing nodules on their host, are in general rather phylogenetically distant from pathogenic microbes[3], and how mutualism emerged in rhizobia remains unclear[4,5]. To our knowledge,

a rhizobial strain harboring features of both pathogenic and nodulation abilities on a single host has not been identified so far.

Like other plants, legumes are challenged by the presence of many microbes including harmful and beneficial ones, and evolution resulted in perception mechanisms that allow plants to recognize friends and foes[6].

Amongst the beneficial microorganisms interacting with legumes, rhizobia can confer a crucial advantage to their hosts when developing on nitrogen-poor substrates. Rhizobia-legumes symbiotic interactions

[1]Université Paris-Saclay, CNRS, INRAE, Université Evry, Institute of Plant Sciences Paris-Saclay, 91190 Gif sur Yvette, France. [2]Université Paris Cité, CNRS, INRAE, Institute of Plant Sciences Paris-Saclay, 91190 Gif sur Yvette, France. [3]Laboratoire des Interactions Plantes Microbes Environnement, Université de Toulouse, INRAE, CNRS, 31326 Castanet-Tolosan, France. [4]DARWIN21, Biological and Environmental Sciences and Engineering Division, King Abdullah University of Science and Technology, Thuwal 23955, Saudi Arabia. [5]Computational Bioscience Research Center, King Abdullah University of Science and Technology, Thuwal, Saudi Arabia. [6]iMEAN, 31077 Toulouse, France. [7]Laboratoire de Recherche en Sciences Végétales, CNRS, UPS, Toulouse INP, Université de Toulouse, Toulouse, France. [8]Metatoul-AgromiX Platform, MetaboHUB, National Infrastructure for Metabolomics and Fluxomics, LRSV, Toulouse, France. [9]Present address: Université Paris-Saclay, INRAE, AgroParisTech, Institute Jean-Pierre Bourgin for Plant Sciences (IJPB), 78000 Versailles, France. [10]Present address: Department of Food, Environmental and Nutritional Sciences (DeFENS), University of Milan, 20133 Milan, Italy. ✉e-mail: benjamin.gourion@cnrs.fr; pascal.ratet@cnrs.fr

result in the formation of specialized root-derived organs, namely nodules, where rhizobia are hosted inside of plant cells. Within nodules, rhizobia can fix atmospheric nitrogen and convert it into ammonium, a form of nitrogen assimilable by the host. As a benefit, rhizobia receives all nutrients from the plants[7].

In agreement with the idea that mutualistic relationships can emerge from pathogenic associations, the interaction between rhizobia and legumes is accompanied by intricate molecular dialogues and subsequent infection processes. These interactions involve various molecular actors, including exopolysaccharides, protein secretion systems and Nod factors, which not only contribute to mutualism but also interfere with the signaling pathways of plant innate immunity[2,8–12].

For many years, nodule inhabitants have been considered as pure cultures of rhizobia because of the tremendous densities of these bacteria within nodules and because of the molecular dialogue that takes place between the plant and the rhizobia in the rhizosphere that leads to specific interactions. However, this view has been drastically challenged and it is now admitted that, in addition to nitrogen-fixing rhizobia, legume nodules host complex and diverse microbial populations called the nodule accessory microbiome[13–15]. The roles of these microorganisms as well as the way they enter nodules remain largely unknown.

In this study, our primary objective was to characterize nodule endophytes, and thus we initially isolated bacterial strains from nodules of *Medicago littoralis* R108 (formerly *Medicago truncatula*

R108) inoculated with soil. Subsequently, we identified a distinctive strain of *Ensifer adhaerens*, referred to as T4, which exhibited divergent interactions with its host depending on the developmental stage of the plant and on the presence or absence of bona fide rhizobia capable of inducing nodules. Notably, we discovered that T4 had the ability to co-occupy nodules induced by other rhizobia, but it can also independently trigger nodule formation. Remarkably, we observed that the inoculation of young seedlings with T4 triggered the unexpected outcome of plant death.

We also showed that the versatile behavior of T4 was not restricted to the *Medicago* genus and that it affected various legume species from the Inverted Repeat-Lacking Clade (IRLC), a monophyletic subclade of the Papilionoideae.

## Results

### *Ensifer adhaerens* T4 has versatile interactions with its host

Bacterial endophytes were isolated from *Medicago littoralis* R108 nodules (R108; formerly *Medicago truncatula* ssp. *tricycla* R108[16,17]) using garden lawn soil as an inoculant (Fig. 1a). Among the isolated bacteria, partial sequencing of *16S rRNA*, *gyrB*, *rpoD* and *recA* suggested that isolates T1, T3 and T4 represented a single *Ensifer adhaerens* (*E. adhaerens*) strain. The strain *E. adhaerens* T4 (T4) was selected for further investigations (Supplementary Data 1). While intending to ascertain, through co- and single inoculations on *M. truncatula* A17 (A17) seedlings, that T4 was a bona fide nodule endophyte, we observed that T4 was capable to both co-infect nodules together with

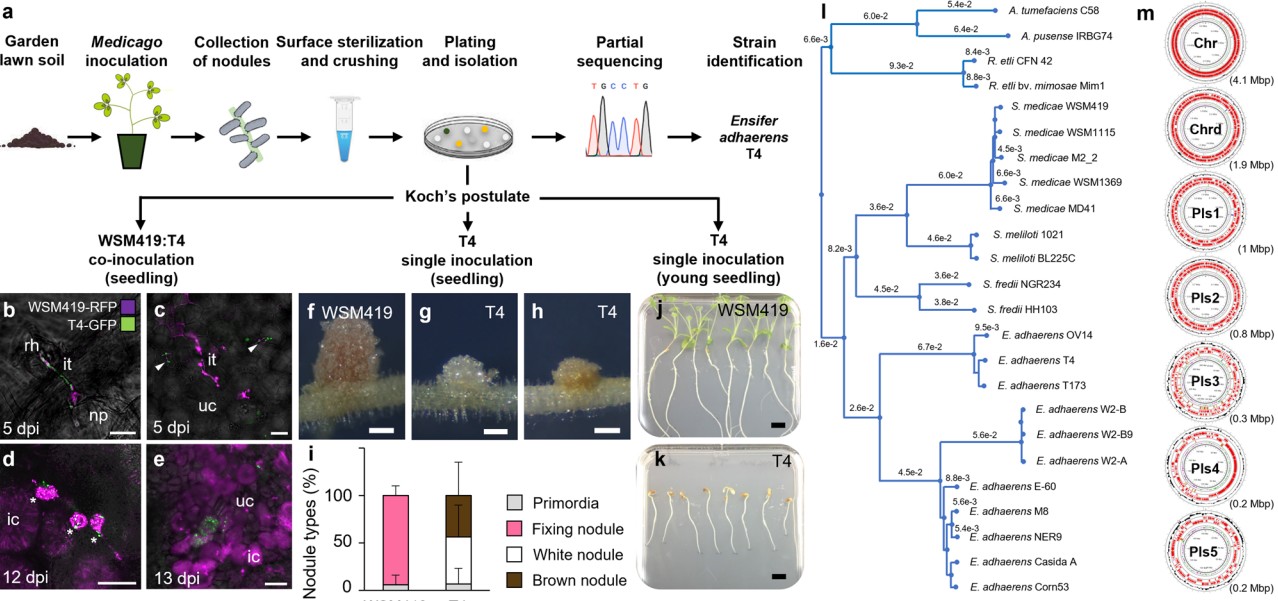

**Fig. 1 | T4 is an *Ensifer adhaerens* strain co-colonizing nodules with effective rhizobia, inducing ineffective nodules by itself and triggering the death of its host. a** Schematic procedure used to trap *Medicago littoralis* R108 (R108) nodule endophytes and to validate the Koch's postulate. **b**–**e** When *Ensifer adhaerens* T4-GFP (T4-GFP, green signal) is co-inoculated with the efficient symbionts *Sinorhizobium medicae* WSM419-RFP (WSM419-RFP, violet signal), T4 behave as a nodule endophyte and co-colonized *M. truncatula* A17 (A17) nodules. **b** A 5-dpi root hair showing both WSM419-RFP and T4-GFP in a single infection thread. **c** WSM419-RFP and T4-GFP in common infection threads (white arrowheads), within a 5-dpi nodule primordium. **d** 12-dpi nodule cells showing the release of WSM419-RFP and T4-GFP bacteroids from infection threads in common infected cells (white asterisks). **e** 13-dpi nodule infected cells containing both WSM419-RFP and T4-GFP. (Scale bars **b**, **c**, **e**, 20 μm; **d**, 50 μm). rh, root hair; it, infection thread; np, nodule primordium; uc, uninfected cell; ic, infected cell; dpi, days post-inoculation. For **b**–**e**, similar results were observed in three independent experiments. **f**–**i** T4 induced, by itself, the formation of nodules on A17 seedlings. **f** 14-dpi

WSM419 nodule. **g, h** 14-dpi T4 nodules. (Scale bar **f**–**h**, 500 μm). For **f**–**h**, similar results were observed in four independent experiments. **i** Percentage of nodule types present on WSM419- and T4-inoculated plants shown as mean percentage ± s.d. (*n* = 24 plants). **j** and **k** T4 triggered the death of A17 young seedlings. WSM419- (**j**) and T4- (**k**) infected plants at 21 dpi. (Scale bars **j** and **k**, 1 cm). **l** Neighbor-joining phylogenetic tree based on whole bacterial genomes including that of T4 and of other strains of the same species, genus or order. **m** Organization of the T4 genome. The T4 genome is composed of seven circular replicons, one chromosome (Chr), one chromid (Chrd), and five plasmids (Pls1 to Pls5). Plasmid sizes are indicated. A remarkable feature of the T4 genome is the heterogeneous density of repeated regions along the different replicons. Especially, Pls3 harbors 27% of repeated sequences, mostly transposase encoding genes, whereas repeated sequences represent less than 3% of the genome in the other replicons (Supplementary Fig. 2; Supplementary Fig. 3). Source data for Fig. 1i are provided in **Source Data** file.

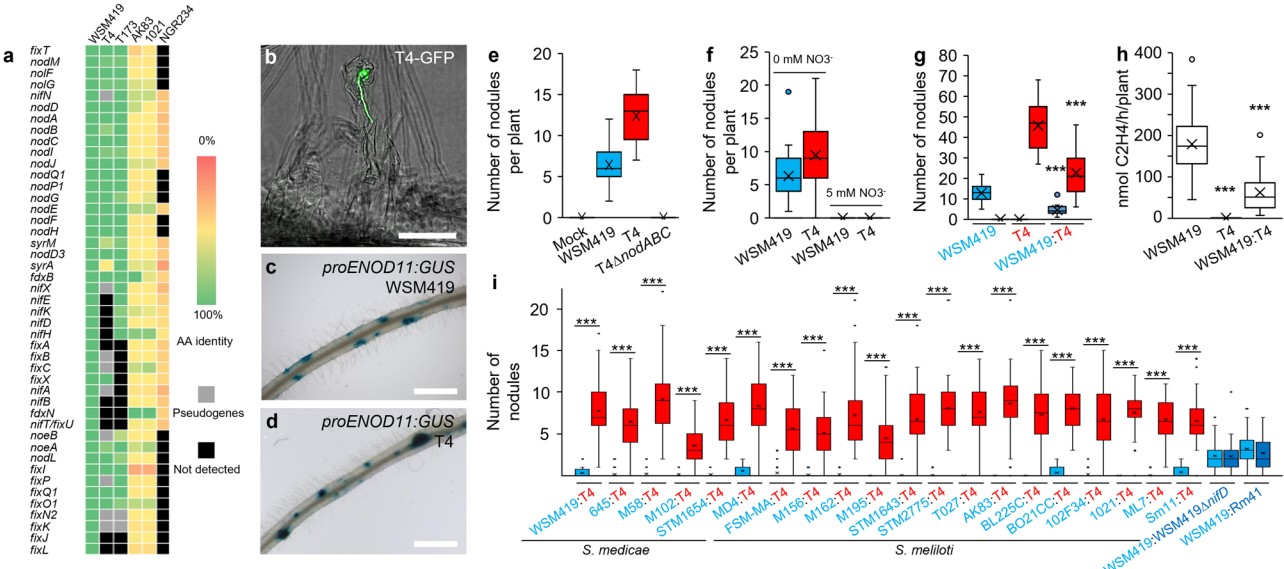

**Fig. 2 | *E. adhaerens* T4 NF-dependent nodulation and T4 competitiveness. a** T4 harbors *nod* genes but lacks core *nif* genes. Heatmap representing the presence/absence of symbiotic genes in T4, T173, *S. meliloti* AK83, *S. meliloti* 1021 and *S. fredii* NGR234 using WSM419 as a reference. The color scale indicates the percentage of identities (amino acid, %) relative to WSM419 reference proteins. For each replicon carrying the nod genes of each strain, detected pseudogene and lacking genes are highlighted in gray and black, respectively. **b–f** The nodulation ability of the T4 strain relied on the production of Nod factors. **b** Confocal imaging of T4-GFP entering root hair through an infection thread. (Scale bars, 50 μm). Confocal imaging experiments consisted of four independent experiments. **c, d** Induction of the Nod factor-reporter construct *proENOD11:gusA* by WSM419 (**c**) and T4 (**d**) in 2-dpi primary root of A17 stable transformants expressing the *gusA* reporter fusion *proENOD11:gusA* (*n* = 12 plants). (Scale bars, 1 mm). **e** T4 *nodABC* genes are required to induce nodule formation. For mock, WSM419, T4 and T4Δ*nodABC*, *n* = 8, 16, 13 and 32 plants. Experiments have been repeated twice. **f** Nitrogen inhibits T4 nodule formation. For WSM419 0 mM $NO_3^-$, T4 0 mM $NO_3^-$, WSM419 5 mM $NO_3^-$, T4 5 mM

$NO_3^-$, *n* = 39, 39, 30 and 12 plants. Experiments have been repeated twice. **g** T4 induced more nodules than WSM419 upon single inoculation and reduced the formation of WSM419 nodules when co-inoculated. Three biological replicates have been performed (*n* = 30 plants). **h** T4 did not fix nitrogen and reduced the nitrogen fixation of plants when co-inoculated with WSM419. Three biological replicates have been performed (*n* = 30 plants). **i** T4 dominated nodule formation over efficient nitrogen-fixing symbionts. Co-inoculation of T4 with nitrogen-fixing *S. medicae* and *S. meliloti* strains. Each boxplot reflects the analysis of 40 plants. **e–i** Plants were analyzed at 21 dpi. For each box-and-whisker plot, the box contains 50% of the data, the bottom and the top of the box represent Q1 and Q3, respectively, the center line indicates the median, the center cross indicates the mean, the whiskers indicate the data that range within 1.5 time the interquartile range and if they exist, outliers are shown. Asterisks indicate significant differences (***, *P* ≤ 0.001; *P* values in **g** = 9.10⁻¹³ and 7.10⁻¹⁰; *P* values in **h** = 3.10⁻¹⁹ and 7.10⁻¹⁰; Two-sided Student's *t*-test). Source data for Fig. 2e-i are provided in **Source Data** file.

the reference strain *Sinorhizobium medicae* WSM419 (WSM419) and to trigger nodules by itself (Fig. 1b–i). Co-inoculation experiments confirmed that T4 was indeed a nodule endophyte since both T4 and WSM419 co-occupied infection threads within root hair and nodule primordium, and that they were found inside common nodule infected cells (Fig. 1b–e). These co-infection events were relatively rare and in cases of co-occurrence, WSM419 was the dominating population in the nodule (Fig. 1e). When T4 was inoculated alone, the induced nodules did not show the characteristic pinkish coloration of nitrogen-fixing nodules (such as those induced by WSM419 alone or co-occupied with T4). Instead, T4 nodules were small, white or brown and without detectable nitrogenase activity, suggesting that T4 nodules were not functional (Fig. 1f–i and Fig. 2h). The ability of T4 to trigger small white and brown nodules was not restricted to A17 since we also observed this trait in other species, including *M. truncatula* F83005.5, *Medicago sativa* cv. salina and *Medicago sativa* cv. Super GRI8 (Supplementary Fig. 1). Methylene blue staining of T4 brown nodules revealed the accumulation of phenolic compounds, which are typically found upon defense reactions in plants (Supplementary Fig. 1).

In addition to these interactions, we found that when T4 was inoculated alone on young germinating seedlings, T4 triggered the death of the host (Fig. 1j, k). Such a particular dual interaction was also observed using *E. adhaerens* strains T1 and T3 (Supplementary Data 1).

To gain insight into the unique behavior of these *E. adhaerens* strains, we sequenced the genome of the T4 strain. Genome clustering analysis confirmed that T4 grouped with other *E. adhaerens* strains and indicated that T4 was closely related to *E. adhaerens* T173, a bacterium

isolated from *Melilotus albus* nodules ([18]; Fig. 1l). The T4 genome was 8,451,442 bp and composed of seven circular replicons, one chromosome (Chr), one chromid (Chrd) and five plasmids, Pls1 to Pls5, ranking the T4 genome among the largest *E. adhaerens* genomes available today (Fig. 1m; Supplementary Fig. 2; Supplementary Data 2; Supplementary Data 3). Interestingly, Pls3 presented typical features of rhizobial symbiotic mobile genetic elements as it carried a remarkably high density of transposase encoding genes as well as genes potentially involved in the synthesis and secretion of Nod factors (NFs; Supplementary Fig. 2; Supplementary Fig. 3).

Thus, our results highlight that T4 is an atypical *E. adhaerens* strain capable of versatile interactions with its host. T4 can behave as a nodule endophyte co-infecting nodules with bona fide rhizobia, it can induce the formation of nodules by itself and it can behave as a pathogen triggering the death of the host.

## The T4 infection is NFs-dependent and results in fix⁻ nodules

Sequence analysis confirmed that the T4 genome harbors the determinants of nodulation on four loci of the T4 Pls3 (Fig. 2a; Supplementary Fig. 3[19]). Blast searches against the NCBI nucleotide database indicated that genes involved in the production, decoration, and secretion of T4 NFs were almost identical to those of T173 and WSM419 (Supplementary Fig. 3). Phylogenetic reconstruction based on *nodABCIJ* genes indicated that T4, T173 and WSM419 genes form a cluster distinct from that of *Sinorhizobium meliloti* strains (Supplementary Fig. 3). The proximity between WSM419 and T4 *nod* genes was further confirmed by comparing the structure of WSM419 and T4 NFs (Supplementary Fig. 4).

Confocal microscopy showed that T4 entered the plant roots through the root hair infection pathway (Fig. 2b). In addition, the transcriptional fusion between the promoter of the early nodulin encoding gene *MtENOD11* and the *GusA* reporter gene was activated during T4 infection, suggesting that a NFs-dependent molecular dialogue and infection process was taking place (Fig. 2c, d). This was further supported by the analysis of a T4Δ*nodABC* deletion mutant unable to elicit nodules (Fig. 2e) and by the fact that the presence of a nitrogen source inhibited the nodulation ability of T4 (Fig. 2f). In agreement with the fix⁻ phenotype of T4, we did not detect the structural genes *nifHDK* encoding the nitrogenase enzymatic complex in the T4 genome (Fig. 2a and h).

T4 is thus a nod⁺ fix⁻ strain harboring core *nod* genes and lacking core *nif* genes. This makes T4 able to infect root hair in a NFs-dependent manner and to induce nodules that are ineffective for nitrogen fixation.

## The T4 strain is a highly competitive parasite

We next compared the ability of T4 to trigger nodules relative to WSM419. Upon single inoculation, we did not observe any difference in the kinetics of nodule induction when comparing T4 and WSM419 (Supplementary Fig. 5). Furthermore, a histological time series analysis of T4 and WSM419 nodule development revealed that cortical divisions initiated synchronously (Supplementary Fig. 7). However, during later stages of nodulation kinetics, we noticed that while WSM419 stopped to initiate nodules, T4 continued to elicit some. Thereby, more nodules accumulated in T4-inoculated plants as compared to WSM419-inoculated ones (Supplementary Fig. 5; Fig. 2g).

Upon T4 and WSM419 co-inoculation, T4 nodules (small white or brown) were easily distinguishable from those of nitrogen-fixing symbionts (big and pink). Interestingly, when co-inoculated with equal bacterial densities, we systematically observed more nodules induced by T4 than by WSM419 (Fig. 2g). In addition, the nitrogenase activity was significantly reduced relative to plants inoculated with WSM419 alone, indicating that T4 affected the interaction between the host and efficient symbionts (Fig. 2h). Under both single and co-inoculation conditions, the presence of T4 significantly reduced A17 leaf biomass by 53% and 42%, respectively, relative to WSM419 single inoculation (Supplementary Fig. 6). Similar results were observed using *M. sativa* cv. salina as a host plant (Supplementary Fig. 6). Moreover, upon T4-WSM419 co-inoculation, we observed a continuous formation of T4 nodules while the formation of WSM419 nodules was drastically impaired (Supplementary Fig. 6). This higher efficiency in triggering nodules was confirmed using multiple lines of *Medicago* (R108, *M. truncatula* Ghor, *M. sativa* cv. G969, cv. WL903, cv. Super GRI8 and cv. salina; Supplementary Fig. 6). In addition, co-inoculation experiments using up to one thousand times more WSM419 cells than T4 ones only resulted in 34% of WSM419 nodules illustrating the high competitiveness of T4 for nodule formation (Supplementary Fig. 6). Such a success for nodulation during competition was not only observed with WSM419 but was systematically encountered with nineteen other *S. medicae* and *S. meliloti* strains, including strains that have originally been isolated from *M. truncatula* (M102, M162, STM1643, T027, MD4, ML7 and 102F34; Fig. 2i). Similar experiments using either a fix⁻ mutant (WSM419Δ*nifD*) or a wild type fix⁻ strain (RM41) instead of T4 during competition with WSM419 did not show such a competitive phenotype, indicating that the T4 inability to fix nitrogen was not the main determinant of its competitiveness (Fig. 2i).

Taken together, these data indicate that T4 is particularly competitive for nodule induction and that the accumulation of T4 fix⁻ nodules instead of nitrogen-fixing ones limits plant development.

## T4 nodules undergo early senescence and defense reactions

A time series histological analysis of T4 and WSM419 nodule development indicated that rapidly after the initiation of nodule organogenesis, the development of T4 nodules stopped, suggesting an arrest of the meristematic activity (Fig. 3a-f; Supplementary Fig. 7). During infection, as for WSM419, T4 reached the symbiotic cells and were internalized, however, within T4 nodules, symbiotic cells showed senescence features (Fig. 3g, i; Supplementary Fig. 7). Indeed, as early as five dpi, 72% of T4-infected cells presented collapsed vacuoles and this phenotype reached 100% of cells by eight dpi while it was not observed for WSM419-infected cells (Fig. 3g-j; Supplementary Fig. 7). In addition, from five dpi, the size of T4 infected and uninfected cells was significantly reduced relative to WSM419 (Fig. 3g-j; Supplementary Fig. 7). In agreement with these senescence features, we also reported the induction of senescence marker genes in T4 nodules, including *MtCYSTEINE PROTEASE2, 3, 5* (*MtCP2, 3, 5*) and *MtNAC969* (Fig. 3k[20,21];). Taken together, these results indicated that T4 nodules prematurely underwent a senescence program.

In agreement with the accumulation of phenolic compounds, suggesting defense reactions in T4 nodules, the expression of ethylene biosynthesis and defense marker genes was induced (Supplementary Fig. 1; Supplementary Fig. 8). In line with the defense and the senescence reactions described above, we also detected the reduced expression of key symbiotic genes required to prevent defense responses and nodule senescence during *Medicago*-rhizobium symbiosis, including *MtSymbiotic CYSTEINE-RICH RECEPTOR-LIKE KINASE* (*MtSymCRK*), *MtDEFECTIVE IN NITROGEN FIXATION2* (*MtDNF2*) and *MtREGULATOR OF SYMBIOSOME DIFFERENTIATION* (*MtRSD*; Fig. 3k[22]). In addition, we reported the non-induction of the *LEGHAEMOGLOBIN1* gene in T4 nodules relative to WSM419 nodules (*MtLEGH1*; Fig. 3k).

We thus concluded that T4 nodules rapidly stopped their development and underwent induced senescence accompanied by defense reactions.

## T4 bacteroids lost viability before morphological differentiation

In agreement with the activation of senescence and defense reactions, live/dead staining of bacteroids revealed that from twelve dpi most intracellular T4 cells had lost their viability (Fig. 3l-u). This result was further confirmed using T4-GFP and WSM419-RFP labeled-strains (Supplementary Fig. 9). Remarkably, even at 21 dpi, while most T4 bacteroids were dead, some alive T4 bacteroids were detected, likely representing freshly released bacteria from infection threads (Fig. 3v, w). Confocal microscopy analyses also showed that T4 bacteroids did not undergo the morphological differentiation typically observed for rhizobia during *Medicago*-rhizobium symbiosis (Fig. 3x-ab[23]). Such a differentiation process is mediated by plant peptides that are maturated by the signal peptidase *MtDEFECTIVE IN NITROGEN FIXATION 1* (*MtDNF1*) and targeted to the symbiosome[24,25]. In the *Mtdnf1-1* mutant background, the loss of viability of T4 was delayed, suggesting a contribution of plant-secreted peptides to the death of T4 bacteroids inside symbiotic cells (Fig. 3ac–aj).

Together, these data showed the premature death of T4 bacteroids prior to the morphological differentiation.

## T4 is pathogenic on juvenile seedlings

The strain T4 triggered plant death upon inoculation on young germinating seedlings (Fig. 1j, k). While inoculation of WSM419 did not alter plant viability, T4 induced disease comparable to that of two well-known *Medicago* pathogens, namely, *Xanthomonas campestris* pv. *campestris* (*Xcc*) and *Xanthomonas euvesicatoria* pv. *alfalfae* (*Xea*; Supplementary Fig. 10).

Interestingly, older A17 were no longer susceptible to T4. To investigate this susceptible-to-resistant shift in A17, we inoculated germinating seedlings from different ages, ranging from zero to six days post-stratification (dps) and monitored their primary root development as a quantitative read-out of survival (Fig. 4a, Supplementary Fig. 11). The strongest deleterious effect was observed when T4 was inoculated from zero to one dps. When inoculated at zero dps,

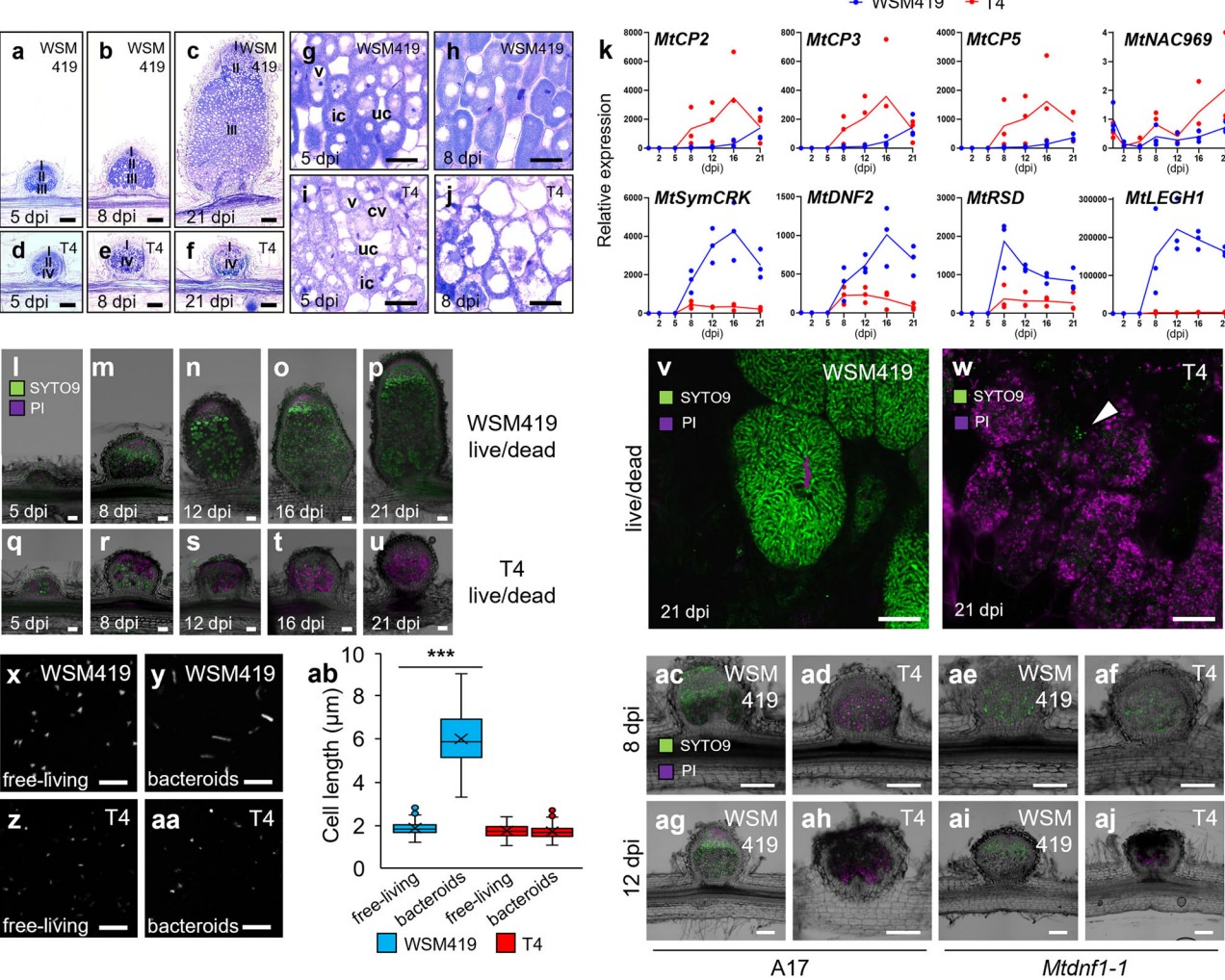

**Fig. 3 | *E. adhaerens* T4 nodules are prematurely stopped in their development and undergo a senescence process. a–f** Histological analysis of WSM419 (**a–c**) and T4 (**d–f**) nodule development. I, meristematic zone; II, infection zone; III, fixation zone and IV, premature senescence zone (IV). **g–j** Histological analysis of WSM419 (**g, h**) and T4 (**i, j**) nodule infected cells. v, vacuole; cv, collapsed vacuole; uc, uninfected cell and ic, infected cell. (Scale bars **a–f**, 200 μm; **g–j**, 40 μm). For **a–j**, similar results were observed using three independent kinetics. **k** RT-qPCR gene expression profiles of senescence marker genes *MtCP2*, *MtCP3*, *MtCP5* and *MtNAC969*, and of symbiotic marker genes *MtSymCRK*, *MtDNF2*, *MtRSD* and *MtLEGH1* along WSM419 (blue curves) and T4 (red curves) nodulation kinetics. Dots represent three independent biological replicates (*n* = 3). Gene accessions are provided (Supplementary Data 4). **l–u** Live/dead staining showing the viability of WSM419 and T4 bacteroids during nodulation. SYTO9 green signals indicate alive bacteroids. PI violet signals indicate dead bacteroids, cell walls, nuclei and nodule meristem. (Scale bars **l–u**, 100 μm). For **l–u**, the kinetics experiment was done once. **v** and **w** Live/dead staining showing WSM419 and T4 bacteroid viability and morphological differentiation within infected cells of 21-dpi nodules. **v** WSM419 nodule-infected cells were densely filled with morphologically differentiated and alive bacteroids. **w** T4 nodule-infected cells were occupied by dead and undifferentiated bacteroids. Some T4 bacteria were alive (Arrowhead). (Scale bars **v** and **w**, 20 μm). **x–ab** Morphological differentiation and cell size of WSM419 and T4 under free-living and bacteroid states. **x–aa** Fluorescence microscopy showing DAPI-stained undifferentiated free-living WSM419 (**x**), differentiated WSM419 bacteroids (**y**), undifferentiated free-living T4 (**z**) and undifferentiated T4 bacteroids (**aa**). (Scale bars **x–aa**, 10 μm). For **x–aa**, pictures were acquired from a single experiment. **ab** Cell length of WSM419 and T4 under free-living and bacteroid states. For each box-and-whisker plot, the box contains 50% of the data, the bottom and the top of the box represent Q1 and Q3, respectively, the center line indicates the median, the center cross indicates the mean, the whiskers indicate the data that range within 1.5 times the interquartile range and if they exist, outliers are shown (*n* = 100 cells). Asterisks indicate a significant difference compared to free-living WSM419 (\*\*\**P* value ≤ 0.001; *P* value in **ab** = 2.10$^{-84}$; Two-sided Student's t-test). **ac–aj** Live/dead staining showing the viability of WSM419 and T4 bacteroids in A17 and *Mtdnf1-1* at 8 and 12 dpi. In *Mtdnf1-1*, the death of T4 bacteroids is delayed compared to the wild type. (Scale bars **ac–aj**, 200 μm). The experiment was performed twice. Source data for Fig. 3k, ab are provided in **Source Data** file.

plant growth stopped early, after three days post-inoculation (dpi). When inoculated at one dps, root development was a bit less impacted and the growth arrest arose later. For two dps inoculation, the negative impact of T4 was still significant but mild and from three to six dps no significant effect was observed (Fig. 4a). We thus observed a susceptible-to-resistant shift of A17 which began after one dps. We also observed that reducing the bacterial concentration of the T4 inoculum attenuated symptoms (Supplementary Fig. 12).

To decipher if the T4 pathogenicity relied on the secretion of specific compounds, we inoculated A17 with the cell-free spent medium of A17 seedling-induced T4 liquid culture. This did not impair plant viability suggesting that T4 bacterial cells are required to trigger pathogenicity (Supplementary Fig. 13). Aside from plant death and root growth arrest, T4-infected symptomatic plants did not harbor morphological changes at the root level, however, plants displayed symptoms on cotyledons (no aperture; Fig. 4b–i). Indeed, seven days after the treatment of susceptible plants (0-1 dps), all mock-inoculated plants displayed open cotyledons with the first leaf coming out while 96% of T4-inoculated plants harbored closed cotyledons indicating that plant development was blocked (Fig. 4b–i). For resistant plants

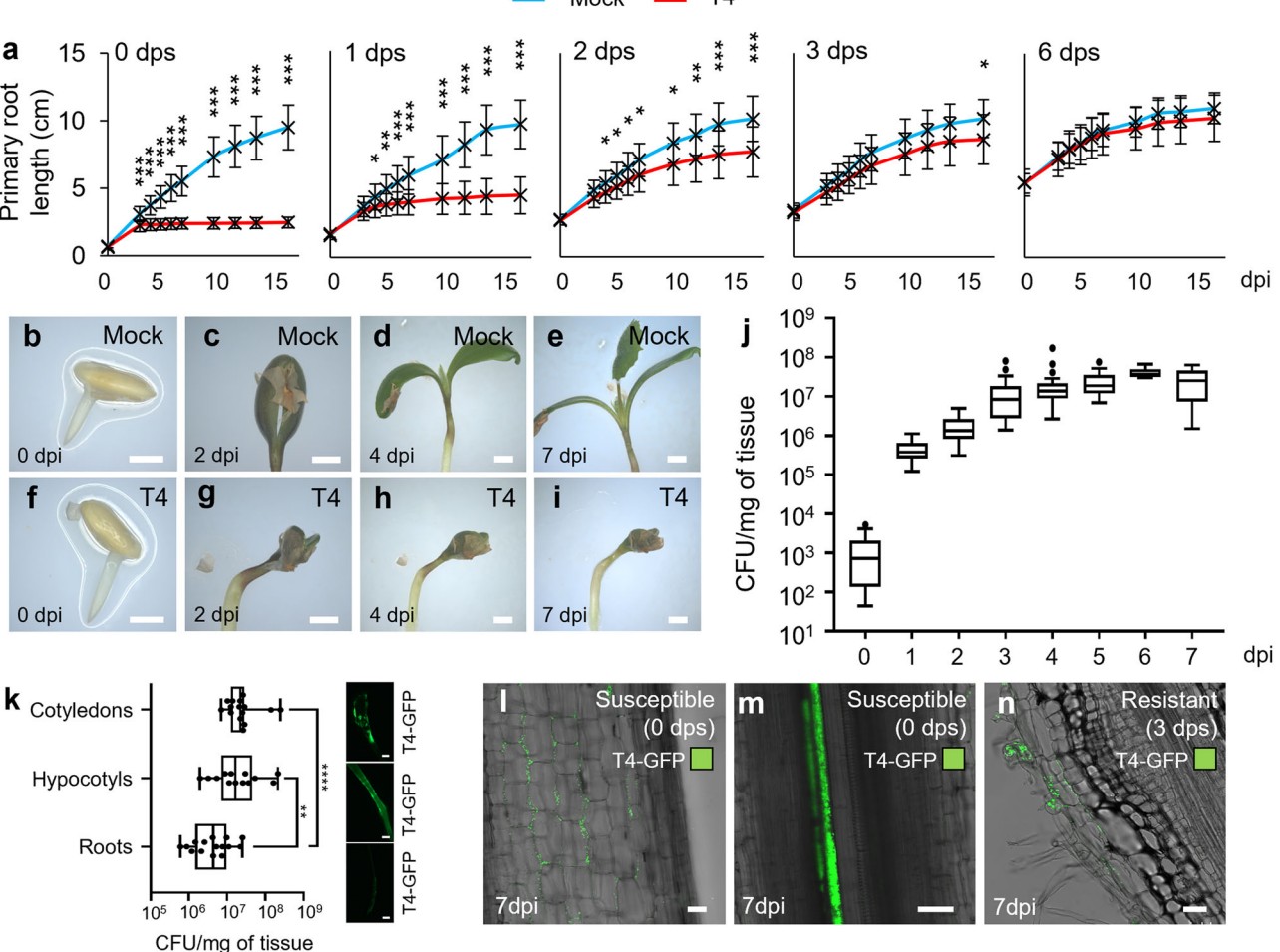

**Fig. 4 | T4 behaves as a virulent pathogens affecting young *M. truncatula* seedlings. a** A17 susceptible-to-resistant shift. Plants were inoculated, or not, with T4 (OD$_{600nm}$: 0.1) at 0, 1, 2, 3 and 6 days post-stratification (dps). Primary roots length was monitored as a plant health proxy. Data represent means ± SEM of three biological replicates (*n* = 60 plants). Asterisks indicate significant differences (**P* ≤ 0.01; ***P* ≤ 0.001; ****P* ≤ 0.0001; Two-sided Student's *t*-test). **b**–**i** T4 triggers the non-aperture of 0-dps A17 cotyledons (*n* = 51 plants). (Scale bars **b**–**i**, 2 mm). **j** Dynamic of T4 proliferation on the whole A17 seedling. For 0, 1, 2, 3, 4, 5, 6 and 7 dpi, *n* = 19, 15, 15, 23, 23, 10, 10 and 5 plants. **k** T4 colonization of susceptible A17 cotyledon, hypocotyl and root at ten dpi. Experiments were performed twice. For cotyledon, hypocotyl and root, *n* = 15, 13 and 15 organs. Asterisks indicate significant differences (***P* ≤ 0.01; ****P* ≤ 0.0001; *P* values = 4.10⁻³ and 4.10⁻⁵; Two-sided Mann and Whitney test). Pictures show corresponding T4-GFP colonizations. (Scale bars **k**, 1 mm). **j** and **k** For each box-and-whisker plot, the box contains 50% of the data, the bottom and the top of the box represent Q1 and Q3, respectively, the center line indicates the median, the center cross indicates the mean, the whiskers indicate the data that range within 1.5 time the interquartile range and if they exist, outliers are shown. **l**–**n** Localization of T4-GFP in susceptible (**l** and **m**) and resistant (**n**) A17 roots. (Scale bars **l**–**n**, 50 μm). The experiment was performed twice and similar results were observed. Source data for Fig. 4a, j, k are provided in **Source Data** file.

(3 dps and after), T4-inoculated plants did not present closed cotyledons anymore. We also observed the rapid proliferation of T4 on susceptible A17 whole seedlings along the days following inoculation (Fig. 4j). In addition, we observed that T4 preferentially colonized A17 cotyledons and hypocotyls rather than roots (Fig. 4k).

At seven dpi, in susceptible plants, we observed the proliferation of T4-GFP inside the root tissues where they colonized the intercellular space of the root cortex and reached the root vasculature (Fig. 4l, m). In resistant plants, the T4-GFP strain displayed a distinct colonization pattern with a reduced presence of bacteria on the root surface and an entry in the plant tissues only through root hair infection threads without any colonization of intercellular spaces or vasculature (Fig. 4n).

Interestingly, the inoculation of mature WSM419-RPF nodules with T4-GFP did not have any impact on nodules and T4 was not detected within nodules. This suggested that nodules do not represent an entry point for T4 (Supplementary Fig. 14).

Taken together, these data indicated that immediately after germination, from zero to one dps, A17 seedlings were susceptible to T4 triggering damping-off symptoms. Later on, between two and three dps, A17 became resistant to T4.

## T4 triggers disease on various IRLC legume species

Besides A17, T4 also triggered disease symptoms leading to plant death on young seedlings of various *Medicago* species and lines including *M. truncatula* Parragio, F83005.5 and Ghor, *M. littoralis* R108, *M. sativa* cv. Super GRI8, WL903, G969, Salina, Oleron. T4 also triggered death of other species from the Inverted Repeat-Lacking Clade (IRLC) including *Melilotus albus, Melilotus officinalis* and *Trigonella calliceras*. However, T4 showed faint to no effect on the viability of *Trifolium repens, Trifolium pratens* and *Trifolium subterraneum, Pisum sativum, Vicia hirsuta, Lens culinaris* or on the phylogenetically more distant legumes *Glycine max, Lotus japonicus* Gifu, *Sesbania rostrata, Astragalus, Galega orientalis* and *Galega officinalis* (Fig. 5a).

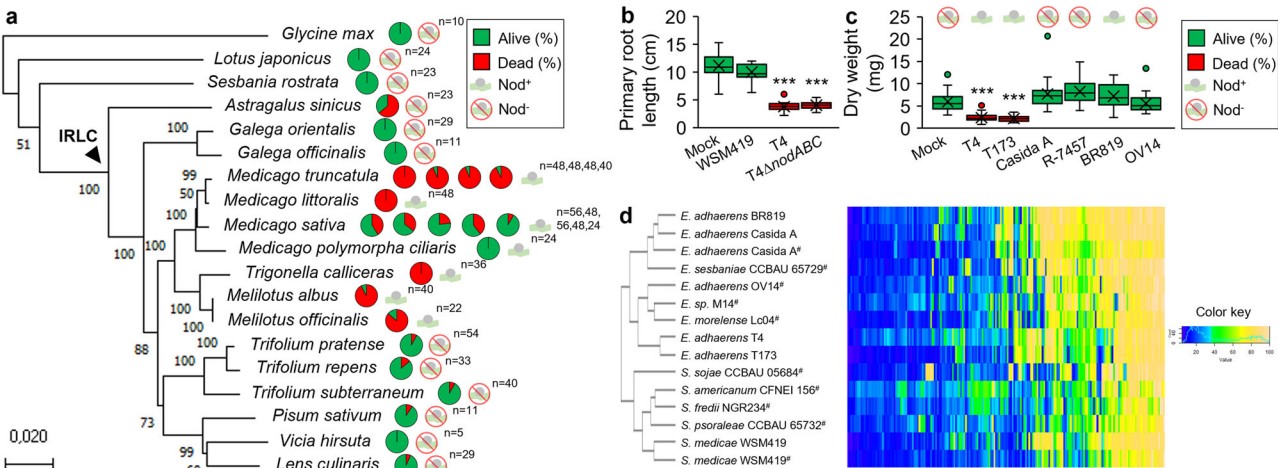

**Fig. 5 | T4 and T173 form a taxonomic cluster of virulent pathogens affecting young seedlings from various IRLC species. a** Pathogenic and nodulation abilities of T4 on legume species. Pie charts represent the percentage of alive and dead plants. For *M. truncatula*, charts correspond to A17, Parragio, F83005.5 and Ghor, and for *M. sativa*, to Super GRI8, WL903, G969, Salina and Oleron. Nod⁺ and Nod⁻ indicate nodulation ability. Maximum Likelihood phylogenetic tree shows the evolutionary history of legumes (Supplementary Data 5). The IRLC is indicated by an arrowhead. **b** The T4 virulence is NFs-independent. Primary root length and mortality at 21 dpi were used as proxy. For mock, WSM419, T4 and T4Δ*nodABC*, *n* = 16, 16, 16 and 32 plants. **c** Evaluation of the virulence of different *E. adhaerens* strains. Plant dry weight and mortality at 16 dpi were used as readouts. For mock, T4, T173, Casida A, R-7457, BR819 and OV14, *n* = 56, 64, 31, 32, 32, 32 and 32 plants.

Nod⁺ and Nod⁻ indicate nodulation ability. **b** and **c** For each box-and-whisker plot, the box contains 50% of the data, the bottom and the top of the box represent Q1 and Q3, respectively, the center line indicates the median, the center cross indicates the mean, the whiskers indicate the data that range within 1.5 time the interquartile range and if they exist, outliers are shown. Asterisks indicate significant differences compared to mock treatments (***$P \leq 0.0001$; *P* values in **b** = $2.10^{-13}$ and $6.10^{-22}$; *P* values in **c** = $2.10^{-22}$ and $2.10^{-15}$; Two-sided Student's *t*-test). **d** Hierarchical clustering analysis of biolog phenotypic microarray datasets. The heatmap represents normalized areas under the curve for carbon usage plates PM01 and PM02 (Supplementary Data 6). #, data retrieved from Fagorzi et al., 2020. Source data for Fig. 5a, b, c are provided in **Source Data** file.

T4 had no pathogenic effect on *Medicago polymorpha ciliaris*, a species on which T4 induced nodules indicating that pathogenicity and ability to nodulate can be uncoupled (Fig. 5a). In agreement and despite that NFs were shown to suppress immunity[2], we did not detect any virulence defect when using a T4Δ*nodABC* mutant unable to produce NFs, demonstrating that NFs were not required for T4 pathogenicity (Fig. 5b).

Our results indicate that T4 can infect a range of legume species belonging to the IRLC, triggering deleterious effects in most of them.

### T4 belongs to a clade containing nodulating-parasitic strains

Although similar in their organization, with large synthenic regions, the genomes of T4 and T173 displayed significant specificities. For instance, 1344 and 1801 genes were specific to T173 and T4, respectively, and the Pls5 was specific to T4 (Supplementary Fig. 15).

As demonstrated above, T4 and T173 strains belonged to the same phylogenetic cluster and were clearly separated from most other *E. adhaerens* sequenced strains (Fig. 1l). In agreement, biolog phenotypic microarrays confirmed that the two strains displayed strong similarities at the phenotypic level (Fig. 5d; Supplementary Fig. 16; Supplementary Fig. 17; Supplementary Fig. 18; Supplementary Data 6).

We evaluated whether T173 and four other *E. adhaerens* strains, namely, Casida A, R-7457, BR819 and OV14[26–31] shared with T4, the dual ability to induce nodules and to trigger plant death, by inoculating young A17 seedlings. Our results showed that T173 was the only other *E. adhaerens* strain capable to both induce nodules and to trigger plant death on A17 seedlings (Fig. 5c). Genomic comparison of the sequenced strains T4, T173, OV14, Casida A and WSM419 indicated that a reduced number of coding sequence families were exclusively shared between T4 and T173 (885 CDS families; Supplementary Fig. 19, Supplementary Data 7). These T4-T173 specific CDS families were specifically enriched on Pls2 and Pls4 suggesting that virulence determinants of strain T4 and T173 might be carried on these plasmids (Supplementary Fig. 19).

Therefore, T4 and T173 represent closely related strains able to induce nodules as well as to behave as parasites on a single host plant.

## Discussion

It has recently been proposed that the symbiotic response was mostly in place in the most recent ancestor of the root nodule symbiosis-forming species more than 90 million years ago[32]. Nevertheless, very little is known about how mutualistic symbionts can emerge from parasitic or commensal associations. The evolutionary trajectory of nodulation has been investigated in the past, notably through the artificial transfer of the nodulation ability towards non-nodulating bacterial species. This has been carried out with success using the phytopathogenic bacteria *Ralstonia solanacearum* on *Mimosa pudica* or using *Agrobacterium tumefaciens* on *Phaseolus vulgaris* and *Leucaena esculenta*[33,34]. In addition, phylogenetic proximity was observed between pathogenic and symbiotic bacteria with sometimes blurry boundaries. For instance, the non-pathogenic strain IRBG74 is genomically very close to pathogenic *Agrobacteria* but it nodulates sesbania[35].

We isolated and characterized an atypical bacterial strain, namely *Ensifer adhaerens* T4 (T4), that naturally displays the ability to trigger drastic disease and to induce nodules on a single legume host species depending on the plant developmental stage. Remarkably, we found that another *E. adhaerens* strain, namely T173[18], also shared such an atypical behavior. Our findings are in agreement with previous research works reporting that T173 elicited large numbers of small white fix⁻ nodules on several legume species including *Medicago sativa*, *Medicago lupulina*, *Melilotus albus* and *Macroptilium atropurpureum*, however, its pathogenic behavior was not reported[18]. In addition, we reported the very high competitiveness of the T4 for nodule formation under competition with known efficient and inefficient symbiotic strains. We demonstrated that such a trait was conserved in T173 (Supplementary Fig. 20).

Recent phylogenetic studies demonstrated that the *Ensifer* (syn. *Sinorhizobium*) genus is actually subdivided into two main clades, a symbiotic clade and a nonsymbiotic clade, containing *Sinorhizobium* strains and *Ensifer* strains, respectively[36,37]. Here, we showed that T4 groups together with *E. adhaerens* species and therefore belongs to the nonsymbiotic clade. To date, the T4 strain represents the third known *Ensifer* species belonging to the nonsymbiotic clade while harboring the core *nodABC* genes, together with T173 and *Ensifer sesbaniae* that harbor *nodABC* and *nifHDK* loci, the latter being capable to fix nitrogen[36,38]. Within the *E. adhaerens* species, although harboring substantial genomic differences, T4 and T173 group together and form a phylogenetic cluster relatively distinct from other sequenced *E. adhaerens* strains, suggesting the existence of a taxonomic group exhibiting such a particular pathogenic/nodulating phenotype. The T4 and T173 closest genome available corresponds to that of *E. adhaerens* OV14, a non-pathogenic bacterium isolated from the rhizosphere of *Brassica napus*, which is used for plant genetic transformation as an alternative to *Agrobacterium*[30,31]. The reliability of such a T4-T173 taxonomic group is supported by the fact that strains T4 and T173 are far from being clonal and that they have been isolated from two distinct continents, in France and in Canada, respectively[18]. The organization of T4 and T173 genomes, especially for replicons carrying *nod* genes, is substantially different suggesting either old and divergent or independent and more recent horizontal acquisitions. Based on sequence analysis (Supplementary Fig. 3), it seems reasonable to speculate that *E. adhaerens* T4 and T173 nodulation genes were acquired from either one, or two distinct *S. medicae* strains. The T4-T173 cluster is also supported by the similarity of metabolic capacities that are shared between T4 and T173.

The symbiotic plasmids of T4 and T173 were remarkably compact (T4, 295 kbp; T173, 204 kbp) compared to the size of symbiotic plasmids from other *S. medicae* or *S. meliloti* species (WSM419, 1245 kbp; SU277, 1024 kbp; WSM1115, 1128 kbp; 1021, 1354 kbp). Furthermore, these two plasmids display a high proportion of repeated regions (11% and 27%, for T173 and T4, respectively) which is a typical feature of instability and suggests an ongoing genetic erosion of those replicons. Nevertheless, until now T4 and T173 symbiotic plasmids have been maintained. The capacity of these atypical *E. adhaerens* strains to form and colonize nodules at high frequency without fixing nitrogen might represent an evolutionary asset allowing the diversification of *E. adhaerens* ecological niches. Indeed, we were able to demonstrate that, as regular nitrogen-fixing symbionts, the T4 strain is also able to resume growth from senescent nodules (Supplementary Fig. 21). However, it should be noted that the ecological success of this clade of nodule-triggering soil bacteria does not only rely on their ability to achieve nodulation and to resume growth post nodule senescence but likely results from multiple factors such as their fitness in the soil in the presence of competitors and/or to their ability to benefit from their host through various mechanisms. Notably, the pathogenic behavior of the T4-T173 clade likely contributes to its ecological success. Moreover, it would be relevant to better determine the geographical distribution of such strains.

The nodulation ability has been independently and frequently lost during evolution indicating an apparent selection against symbiosis[39]. Such mutualism breakdowns have been proposed to be due to the spread of non-fixing cheater symbionts exploiting the benefit of the host association without returning any benefit to the host[1,39]. The discovery of the T4-T173 clade and its unexpected nodulating pathogenic strains suggests that the trade-off between cost and benefit in nodulation can be even more unbalanced than when considering only non-fixing rhizobia. Such a clade might contribute to the selective pressure acting against symbiosis in legumes.

It has been proposed that cheaters and mutualists often coexist because of tradeoffs between presence of cheaters, plant benefits from mutualists and costly induction of defenses[40]. When it behaves as an endophyte, the T4 strain follows the typical features of a cheating organism. Except that the T4 uses its own NFs to enter the nodule in contrast to other nodule endophytes hijacking symbiont's NFs signaling[41]. In agreement with our observations, it has also been shown that cheaters are rather present at low levels within *Medicago sativa* nodules mixed-infected with mutualists[42]. The presence of cheaters is likely masked by the dominant mutualists and they thus escape sanction. In addition, it has been reported that non-rhizobial nod⁻ and/or fix⁻ nodule endophytes better survive post-nodulation than mutualistic rhizobia[43]. This reflects that, below a certain threshold, cheaters thrive within nodules and that nodules represent a niche for the proliferation of non-beneficial bacteria.

Our study reports the isolation of a unique clade of parasitic microbes harboring and using NFs. Hence, the atypical behavior of T4 renders this strain very valuable as a tool to better understand the molecular frontiers existing between parasites and mutualists, the evolutionary trajectory of legume-rhizobium mutualistic interactions as well as transition mechanisms occurring along the parasitism-mutualism *continuum*. In addition, T4 can also be used to better understand the genetic control of nodule immunity and chronic infection during legume-rhizobium symbiosis.

## Methods

### Trapping, isolation, DNA extraction and identification of nodule endophytes

Wild type *M. littoralis* R108 seedlings (Formerly *M. truncatula ssp. tricycla*[16,17] were grown in 1.5 L pots containing a mixture of sterile sand-perlite (1:3, v:v) and inoculated with 50 mL of non-sterile garden lawn soil collected in Limours-en-Hurepoix, France, N 48.649137°, E 2.069144°. 30-day-old nodulated roots were harvested, washed under tap water to remove adhering soil particles, sterilized with NaOCl (6%) for 5 min and washed three times with sterile water. Pink (nitrogen-fixing) as well as white and brown (non-fixing) nodules were collected, pooled and crushed in 1.5 mL tubes containing 200 μL of YEM medium[44]. Nodule extracts were plated on YEM agar medium and incubated for 48 h at 30 °C. Nodule endophyte colonies were isolated according to their color, shape, shininess, exopolysaccharide secretion and antibiotic resistance. Bacterial DNA was extracted using NucleoSpin Microbial DNA Mini kit according to the manufacturer's recommendation (https://www.mn-net.com). Partial *16S rRNA*, *gyrB*, *rpoD* and *recA* encoding sequences were amplified by PCR using Q5 High-Fidelity DNA Polymerase according to the manufacturer's recommendation (https://www.neb-online.fr/). PCR products were sequenced (www.eurofinsgenomics.eu) and blast on NCBI for sequence homology (https://blast.ncbi.nlm.nih.gov/). Primers used for PCR and sequencing are provided in (Supplementary Data 4).

### Plant material and growth conditions

Experiments were essentially performed using *Medicago truncatula ssp. truncatula* ecotype Jemalong A17[45]. The A17 transgenic derivatives, A17 *proENOD11::gusA*[46] and A17 *Mtdnf1-1*[24] were also used. Other legume species were used, including *Glycine max, Lotus japonicus* Gifu, *Sesbania rostrata, Astragalus sp., Galega orientalis, Galega officinalis, Medicago truncatula* (Parragio, F83005.5, Ghor)*, Medicago littoralis* R108, *Medicago sativa* (WL903, G969, Salina, Oleron and Super GRI8), *Medicago polymorpha ciliaris, Trigonella calliceras, Melilotus albus, Melilotus officinalis, Trifolium pratense, Trifolium repens, Trifolium subterraneum, Pisum sativum* var. caméor, *Vicia hirsuta* and *Lens culinaris*. Seeds were scarified with sandpaper and surface-sterilized with NaOCl (1.5 g of active chlorine per 1 L of water) supplemented with one droplet of liquid soap for 30 min. Three washes were performed with sterile water. Seeds were stratified for two days at 4 °C under darkness on water agar plates and then transferred for 22 h (for pathogenic assay) or for 70 h (for symbiotic assay) at 24 °C under darkness for germination. Seedlings were grown in vitro on Buffered Nodulation Media (BNM[47]) or in 1.5 L pots containing a mixture of sand-perlite (1:3,

v:v). Plants were grown in controlled environmental chambers under 16 h: 8 h, light: dark, 24 °C: 24 °C, day: night, 60% relative humidity and 200 µE light intensity. Plants grown in sand-perlite mixture were watered with 1 g/L of nitrogen-free nutritive solution (NPK 0-15-40; Plantprod).

## Microbial materials and growth conditions

*Ensifer adhaerens, Sinorhizobium medicae, Sinorhizobium meliloti, Xanthomonas* sp. and *Escherichia coli* strains physically used in this study are provided in Supplementary Data 8.

**Solid culture of rhizobia.** Strains were grown on YEB plates for 48 h at 30 °C[48].

**Liquid culture of rhizobia.** Bacteria were grown in 20 mL of YEB for 48 h at 30 °C.

**Preparation of rhizobia and *Xanthomonas* suspension.** Liquid cultures were centrifuged for 20 min at 3200 × *g*, washed twice and adjusted at $OD_{600\,nm}$: 0.1. For in vitro assays, 1 mL of bacterial suspensions was used per plate containing eight seedlings. For assay in pots containing sand:perlite, 50 mL of bacterial suspension was used per pot containing five seedlings.

## Symbiotic assays, co-inoculation assays and pathogenic assays

**Symbiotic assays.** 22 h post-stratification, seedlings were transferred on BNM plates and 48 h after transfer, seedlings were root-inoculated with 1 mL of a bacterial suspension at $OD_{600\,nm}$: 0.1 (For symbiotic assays, seedlings were inoculated 70 h post-stratification).

**Co-inoculation assays.** Co-inoculation assays were performed as described for symbiotic assays with a mixture of two bacterial suspensions (1:1, v:v) adjusted to final $OD_{600\,nm}$: 0.1 each.

**Pathogenic assays.** A total of 22 h post-stratification, whole seedlings were infected by dipping in bacterial suspension for 1 h and next transferred to BNM plate. (For pathogenic assays, seedlings were inoculated 22 h post-stratification). For both WSM419 and T4, an $OD_{600\,nm}$ of 0.1 corresponded to a concentration of $10^8$ cfu/mL as determined with a spiral plater (easySpiral; www.interscience.com).

## WSM419 and T4 mutant constructions

**T4-GFP.** Spontaneous rifampicin-resistant *E. adhaerens* T4 mutant was first generated[49]. *E. adhaerens* T4 was then tagged on the chromosome using pBK-miniTn7-gfp3 construct[50]. Transformation was done by conjugation using *E. coli* SM10λpir[51] harboring the helper plasmid pUX-BF13[52] and the mobilizer plasmid pRK600[53]. *E. adhaerens* T4-GFP derivatives were screened for GFP fluorescence and genotyped by *16S rRNA* gene sequencing. In this strain derivative, the insertion was checked by PCR and sequencing. The insertion is located downstream the *nodM* locus.

**T4ΔnodABC.** Flanking regions of *nodABC* were PCR amplified using Phusion Taq polymerase, OCB2262/OCB2263 and OCB2264/2265 as primers and T4 genomic DNA as template. PCR products were cloned into pJET1-2 (Thermo Scientific™) and then subsequently juxtaposed as *Sac*I-*Nco*I and *Nco*I-*Sal*I fragments into *Sal*I-*Sac*I-digested pJQ200mp19[54] giving pLS368-1. The absence of mutation in the construct was checked by DNA sequencing. Plasmid was introduced in T4 by triparental mating using pRK600 as a helper[53]. Single-crossover genomic integration of pLS368-1 was generated by selecting for gentamycin (Gm) resistance. The resulting strain was then propagated in the absence of antibiotic and cells having lost the plasmid by a second recombination event were selected by plating on TYC supplemented with 5% sucrose (Suc). $Suc^R$ $Gm^S$ colonies were screened by PCR analysis using OCB2266/OCB2267.

**WSM419ΔnifD.** Flanking regions of *nifD* were PCR amplified using the primers OCB1684/OCB1685 and OCB1682/OCB1683 and WSM419 genomic DNA as template. PCR products were cloned into pGEM-T (Promega™) and then subsequently juxtaposed as *Sal*I-*Bam*HI and *Bam*HI-*Sac*I fragments into *Sal*I-*Sac*I-digested pJQ200mp19[54] giving pLS296-1. The absence of mutation in the construct was checked by DNA sequencing. Plasmid was introduced in WSM419 by triparental mating using the helper plasmid pRK2013[55]. Single-crossover genomic integration of pLS296-1 was generated by selecting for gentamycin (Gm) resistance. The resulting strain was propagated in the absence of antibiotic and cells having lost the plasmid by a second recombination event were selected by plating on TYC supplemented with 5% sucrose (Suc). $Suc^R$ $Gm^S$ colonies were screened by PCR analysis using OCB1866/OCB1867. Primers used to construct bacterial mutants are provided in Supplementary Data 4.

## Acetylene reduction assay

Acetylene reduction assays were carried out using a protocol modified from ref. 56. Briefly, a 21-dpi-nodulated-root system from a single plant was placed in a 21-mL glass vial sealed with a rubber septum in the presence of 200 µL of water. 500 µL of acetylene gas was injected into each vial and a 2-h incubation was performed. For each sample, 1 mL of gas was injected for analysis. Ethylene production was measured by gas chromatography using a gas chromatograph 7820 A (Agilent Technologies) equipped with a GS-Alumina column (50 m × 0.53 mm). $H_2$ and $N_2$ were used as carrier and makeup gases, respectively. Column temperature and gas flow were set at 120 °C and 7.5 mL/min, respectively.

## Microscopy and sample preparation

**Description of technovit samples.** Zero dpi samples, 1 cm of primary root segment was collected 1 cm above the root apex. two dpi samples, 1 cm of primary root segment was taken above the root apex marked at zero dpi (this position has been chosen to maximize the recovery of nodule organogenesis events, according to Shen et al., 2019 showing the maximum response to LCO in the first cm of root above the root apex). 5 dpi samples, nodule primordia. 8, 12, 16, and 21 dpi samples, nodules.

**Technovit resin inclusion.** Samples were fixed for 30 min in 0.05 M sodium cacodylate buffer, pH 7, 1% (v/v) glutaraldehyde, and 4% (v/v) formaldehyde under vacuum (~500 mm Hg), incubated overnight at 4 °C and washed two times for 30 min with sodium cacodylate buffer. Once dehydrated by successive ethanol bath series (10%, 30%, 50%, 70%, 90%, 100%, 100%, 100%, 1 h each), ethanol was progressively replaced by Technovit 7100 (www.kulzer-technik.com) using ethanol:Technovit solutions [(v/v), 3:1, 1:1, 1:3, 0:3, 0:3, 0:3, 1 h each] at 4 °C and under agitation. Samples were included in Technovit resin using Teflon Histoform S embedding molds (Heraeus Kulzer). 5-µm-thick sections were carried out using an RM2155 microtome (www.leica-microsystems.com) and a TC-65 tungsten carbide blade (www.leica-microsystems.com). Samples were stained for 10 min in Toluidine Blue 0.02% (w/v).

**Vibratome semi-thin sections.** Nodule samples were embedded in agarose (6%, w/v) and sectioned using vibratome (VT1200S; www.leica-microsystems.com). The vibratome was set up as follows: speed, 0.60 mm/s; amplitude, 2.55 mm; thickness, 60 µm and continuous mode. Sections were kept in Tris-HCl, 50 mM pH: 7.2 for subsequent analysis.

**LIVE/DEAD staining.** Nodule sections were stained for 20 min under darkness using the LIVE/DEAD BacLight Bacterial Viability Kit (SYTO9, 3.34 µM and PI, 20 µM in Tris-HCl, 50 mM pH 7.2 (www.thermofisher.com). Sections were washed in Tris-HCl, 50 mM pH 7.2 prior observation.

**DAPI staining.** Free-living bacteria and bacteroids from crushed nodules were stained for 10 min with DAPI (4, 6-diamidino-2-phenylindole) 50 μg/mL.

**Confocal laser scanning microscopy.** Fluorescent signals were detected by confocal laser scanning microscopy (LSM880; www.zeiss.fr). Images were acquired and processed using ZEN2.3 lite software (www.zeiss.fr).

**Fluorescent bacteria.** Fluorescent signals from *S. medicae* WSM419-RFP (excitation wavelength, 561 nm; detection wavelength, 606–633 nm) and *E. adhaerens* T4-GFP (excitation wavelength, 488 nm; detection wavelength, 498-567 nm).

**LIVE/DEAD.** Fluorescent signals from alive SYTO9-stained rhizobia (excitation wavelength, 488 nm; detection wavelength, 501–559 nm) and from dead IP-stained rhizobia (excitation wavelength, 561 nm; detection wavelength, 606–633 nm).

**DAPI.** Fluorescent signals from DAPI-stained rhizobia (excitation wavelength, 405 nm; detection wavelength, 425–514 nm).

**Analysis of light microscopy pictures.** Pictures were treated with ImageJ software or observed and captured thanks to a microscope BX53 (OLYMPUS) and the cellSens Standard software (OLYMPUS).

## Promoter:GUS gene expression pattern

Histochemical GUS staining was performed as described in ref. 57. 2-dpi *M. truncatula* A17 transgenic plants expressing the *proENOD11::gusA* reporter construct were inoculated with WSM419 or T4[46]. Root samples were vacuum infiltrated for 30 min (~500 mm Hg) in X-gluc staining buffer (50 mM phosphate buffer (pH 7.2), 1 mM potassium ferricyanide, 1 mM potassium ferrocyanide, 0.1% (w/v) SDS, 1 mM EDTA and 1.25 mM 5-bromo-4-chloro-3-indolyl-beta-d-GlcA containing cyclohexylammonium salts) and incubated overnight at 37 °C under darkness. Samples were fixed in 50 mM phosphate buffer (pH 7.2), 1% (v/v) glutaraldehyde and 4% (v/v) formaldehyde for 15 min under vacuum (~500 mm Hg). Pictures were acquired using a stereomicroscope Stemi 305 (ZEISS).

## RT-qPCR gene expression profiling along the nodulation kinetics

**Description of samples.** 0- and 2-dpi samples, primary root lacking 0.5 cm of root below hypocotyl and 0.5 cm of root above root apex. 5-dpi samples, primary root holding nodule primordia, lacking 0.5 cm of root below hypocotyl and 0.5 cm of root above root apex. 8-dpi samples, nodules with 0.5 cm of subtenting root. 12-, 16- and 21-dpi samples, nodule with minimum subtending root.

**RNA extraction.** Total RNA extractions were performed from frozen tissues using TRIzol reagent (Ambion). RNA samples were treated with the TURBO DNA-free Kit (Ambion) according to the manufacturer's recommendations.

**Reverse transcription.** Full-length cDNA were synthesized from 800 ng of total ARN using the SuperScript II Reverse Transcriptase kit (Invitrogen) in presence of Ribolock RNase Inhibitor (Thermo Scientific).

**RT-qPCR analysis.** RT-qPCR was performed on five times diluted cDNA using LightCycler FastStart DNA Master SYBR Green I kit and a LightCycler 480 II according to the manufacturer's instructions (Roche). Cycle threshold and primer specificities were determined with the LightCycler 480 software release 1.5.0 SP4. Primer efficiencies were calculated with LinReg PCR: Analysis of Real-Time PCR Data,

version 2016.1. *MtACT11* and *MtRNA RECOGNITION MOTIF* reference genes were used for gene expression normalization. Information concerning primers used for RT-qPCR gene expression analyses are provided in Supplementary Data 4.

## T4 colonization analysis

At 10 dpi, T4-inoculated A17 plantlets were collected and rinsed twice with sterile water for 15 seconds under gentle agitation. Cotyledon, hypocotyl and root organs were separated using a sterile razor blade. The fresh weight of individual organs was determined before grinding the material with two mm diameter glass beads in 600 μL of sterile water using a Fastprep-96 (MP biomedicals) for 2.5 min at 1800 rpm. Suspensions were diluted in sterile water and 10 μL of each dilution were spotted on solid TY medium supplemented with kanamycin (50 μg/mL). Plates were incubated for two days at 28 °C before colony counting.

## Bacterial genomic DNA extraction, PacBio library preparation and genome sequencing

T4 genome sequencing was performed at King Abdullah University of Science and Technology (KAUST, Saudi Arabia). Fresh and pure bacterial culture was used for total genomic DNA extraction using Sigma's GenElute bacterial genomic DNA kit (Sigma Aldrich, Germany) following the manufacturer's protocol. DNA quality and quantity was assessed by using NanoDrop 2000 (Thermo Fisher Scientific, USA) and Qubit dsDNA BR assay kit (Thermo Fisher Scientific, USA). DNA was size selected to 10 kb using the BluePippin™ Size-Selection System (Sage Science, USA), following the High-Pass™ DNA Size Selection of ~20 kb SMRTbell™ Templates manual. The SMRTbell™ template library was prepared according to the instructions from Pacific Biosciences's "Procedure & Checklist - 20 kb Template Preparation using BluePippin™ Size-Selection System" guide. The SMRT cells were run at the KAUST Bioscience Core Labs on the PacBio *RSII* (Pacific Biosciences, USA) sequencing platform using P6-C4 chemistry.

## Genome assembly and annotation

PacBio reads were assembled into seven circular contigs by using Flye v.2.9.1 (https://github.com/fenderglass/Flye[58]) with default parameters. Circularization of contigs was automatically performed by Flye, circularity has been checked by aligning contigs against themselves using Gepard[59]. OriC sites were identified using Ori-Finder 2 and replicons were restarted according to OriC sites when detected[60]. Genome annotation was conducted using the Microscope platform interface (https://mage.genoscope.cns.fr[61]).

## Phylogenetic reconstruction of bacteria based on whole genomes

The phylogenetic tree was generated using the Microscope platform interface (https://mage.genoscope.cns.fr[61]). The genomic similarity was estimated using Mash software computing a distance between two genomes (https://github.com/marbl/Mash). This distance is correlated to the ANI like: D ≈ 1-ANI. From all the pairwise distances of the genomes set, a tree is constructed dynamically using the neighbor-joining javascript package (https://www.npmjs.com/package/neighbor-joining). The tree displays clustering annotations. The clustering has been computed from all-pairs distances ≤ 0.06 (≈ 94% ANI) that correspond to the ANI standard to define a species group using the Louvain Community Detection (https://github.com/taynaud/python-louvain).

## Phylogenetic reconstruction of legume species based on *matK* sequences

The evolutionary history of legume species was inferred by using the Maximum Likelihood method and Tamura-Nei model[62]. Chloroplastic *matK* sequences, retrieved from NCBI, have been used (Supplementary

Data 5). Log likelihood: -10895,59. The percentage of trees in which the associated taxa clustered together is shown next to the branches. Neighbor-Join and BioNJ algorithms were applied to a matrix of pairwise distances estimated using the Tamura-Nei model. The scale corresponds to the number of substitutions per site. Codon positions included were 1$^{st}$+2$^{nd}$+3$^{rd}$+Noncoding. 2597 positions were considered in the final dataset. The phylogenetic reconstruction has been done using MEGA X[63].

## Phenotype Microarray analysis

The Phenotype Microarrays data were analyzed using the R package OPM version 1.0.6[64]. For the original data generated in this study, means of the area under the curve for the replicates of each strain were calculated using OPM. For the data reused from[36] containing only one replicate, the area under the curve was calculated using OPM. Data were normalized to 100 relative to the maximal value found in each microplate (Supplementary Data 6). The heatmap was drawn using the heatmap function of the OPM package with default clustering method for the combined PM01 and PM02 plates.

## Reporting summary

Further information on research design is available in the Nature Portfolio Reporting Summary linked to this article.

## Data availability

All data related to this study are included in the manuscript, in Supplementary information and in Supplementary Data. The genomic data of the *E. adhaerens* T4 strain are accessible on MicroScope - Microbial Genome Annotation & Analysis Platform (https://mage.genoscope.cns.fr) as well as on NCBI under the BioProject ID: PRJNA1066792. All genetic materials used in this study are available on request to Pascal Ratet or Benjamin Gourion (pascal.ratet@cnrs.fr; benjamin.gourion@cnrs.fr). The *E. adhaerens* T4 strain is also available at the CIRM-CFBP French Collection for Plant Associated Bacteria (https://cirm-cfbp.fr/) under the accession number CFBP 9181. Source data are provided with this paper.

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

## Acknowledgements

This work has been supported by the grant ANR-21-CE20-0016-01-PATHOSYM attributed by the French National Research Agency to **PR**. **PR** is supported by the French Laboratory of Excellence "SPS" ANR-10-LABX-0040-SPS, ANR-17-EUR-0007 and EUR SPS-GSR which is managed by the French National Research Agency under the program "Investissements d'avenir" ANR-11-IDEX-0003-02. **BG** and **FV** are supported by the French Laboratory of Excellence "TULIP" ANR-10-LABX-41 and ANR-11-IDEX-0002-02. **VG** is supported by the IdEX Université de Paris ANR-18-IDEX-0001. **FPP** and **FM** are from MetaToul platform (Toulouse metabolomics & fluxomics facilities, www.metatoul.fr) which is part of the French National Infrastructure for Metabolomics and Fluxomics MetaboHUB-ANR-11-INBS-0010. **ER** was supported by the grant FP7-609398 attributed in the frame of the Agreenskills+ postdoctoral fellowship programme which depends on the EU's Seventh Framework Programme. The Laboratory of Bioinformatics Analyses for Genomics and Metabolism - LABGeM (CEA/Genoscope & CNRS UMR8030), France Génomique and the French Bioinformatics Institute (funded as part of the "Investissement d'Avenir" program managed by Agence Nationale pour la Recherche, ANR-10-INBS-09 and ANR-11-INBS-

0013) are acknowledged for their support within the MicroScope annotation platform. We thank Sylvie Fournier and Guillaume Marti from the Metatoul platform for their help in the NFs characterization. We thank Dr. Julie Cullimore (LIPME, France), Dr. Benoit Alunni (I2BC, France), Dr. Peter Mergaert (I2BC, France) and Dr. Ouartsi Akilafor (Université Badji Mokhtar-Annaba, Algeria) for providing most rhizobial strains used for T4-competitiveness assays. We thank Dr. Gabriella Endré (Szeged Research Centre for Biology, Hungary) for providing WSM419-RFP and WSM419-GFP derivatives. We thank Dr. Eden S. P. Bromfield (Ottawa research and development centre, Canada) for providing the *E. adhaerens* T173 strain. We thank Dr. Anne Willems (Ghent University, Belgium) for providing *E. adhaerens* Casida A, R-7457 and BR819 strains from the BCCM/LMG collection (https://bccm.belspo.be/). We thank Dr. Ewen Mullins (Teagasc Crops Research Centre, Ireland) for providing the *E. adhaerens* OV14 strain. We thank again Dr. Benoit Alunni and Dr. Peter Mergaert for providing seeds of legumes from the IRLC. We thank Dr. George C. diCenzo (Queen's University, Canada) for providing published biolog datasets.

## Author contributions

P.R., B.G., F.V., and K.M. designed research; K.M., S.M., T.F., L.S., E.A.S., I.P., M.M.S., A.A.E., S.B., A.J., E.R., V.P.P., F.M., G.B., C.G., and B.G. performed research; K.M., T.F., L.S., E.A.S., I.P., M.M.S., A.A.E., S.B., A.J., E.R., R.F., V.P.P., H.H., V.G., R.P., F.V., B.G., and P.R. analyzed data; K.M. and B.G. wrote the manuscript.

## Competing interests

The authors declare no competing interests
