## [Peer Review File · Nature Communications]

REVIEWER COMMENTS

Reviewer #1 (Remarks to the Author):

The manuscript by Magne et al. describe a novel strains of a root nodule endophytic bacterium able to colonize in a pseudorhizobial form root nodules and induce their formation on some host legumes. A detailed panel of biochemical, genomic and morphological analyses have been performed and clearly reported and commented.

This discovery is giving momentum to the study of the evolution of symbiotic nitrogen fixation and deserve the merit to be published.

However, some minor points should be fixed and a more precise comment on the ecological context this strain could thrive should be given.

Specific points follow:

Line 71. Some check of English could be perfoemd. E.g. a comma after “Despite that”.

Line 113-114. Here you may better report previous findings and comments in the discussion section under the light of advantages of cheating and balancing selection for symbiosis vs. cheating. See for instance: <https://doi.org/10.1111/j.1574-6941.2007.00424.x> <https://doi.org/10.1111/ele.13247> <https://doi.org/10.3389/fpls.2016.00835>

Line 167. Is there one of the replicon a chromid? In other words, does it contains par system?

Figure 4 Q. The clade of *Ensifer sojae*-*E. medicae* should be named as *Sinorhizobium*. Please use the term *Sinorhizobium* along the text for members of this clade since *Ensifer* is not a valid name anymore for that clade (see you ref. n. 26 and paper in *IJSEM* <https://doi.org/10.1099/ijsem.0.005243>)

Line 548. Genome data availability must be reported as bioproject or similar on international databases (e.g. genbank or ENA)

Line 549. Since the importance of such a strain a deposit on a publicly accessible strain repository (e.g. DSMZ) must be done

Fig. S16. A more readable representation could be better, why not to use a heatmap or similar?

Table S2. Please do not copy and past and excel datasheet without any legend and with some columns names partially masked!

Table S6. This table is mostly unreadable. The same for Table S7.

Reviewer #2 (Remarks to the Author):

This is a very thorough and interesting piece of work that suggests that T4 is a pathogenic strain of *Ensifer* that has nod but not nif genes. It would appear the acquisition of nod genes allows T4 to form aborted infections on *Medicago* spp. I have specific questions below but the thrust of my comments is about the effects on population size of T4 in the possession of nod genes, but in the absence of nif genes and N₂ fixation and mutualistic cooperation with the plant. My guess is that T4 is quickly shut down by the plant in nodules and therefore is released from nodules in much smaller numbers (if at all) than a symbiotically effective strain. Just as we see for poorly effective strains of *Rhizobia* (that possess nod and nif genes), sanctioning by the plant means these cheating strategies may work up to a point but such strains are kept to a low frequency. However, this of course is my speculation because these experiments are not shown in the work so far.

-It is not clear where T4 and indeed T1 and T3 were isolated from. Were they isolated as small white or brown nodules or from coinfecting nodules? I suspect individual small white/brown nodules as these would be easy to find but I am not sure from the paper. This leads directly to my next question which is what percentage of nodules in the original soil inoculum had these ineffective T strains? This is actually a crucial ecological question as such parasitic/pathogenic strains would be very harmful to *Medicago* spp. if present at high levels in soil.

-A question of major importance is how many T4 bacteria regrow from the ineffective nodules over a time course and crucially how does this compare to an effective strain? The time course also needs to be run for a reasonable long time such as 6-8 weeks to replicate senescence. The reason for this question is that for this to be an evolutionary successful strategy T4 must get released from nodules and be able to regrow. I would expect there to be orders of magnitude more rhizobia released from effective nodules (e.g. WSM419 infected) than T4 infected nodules. This also relates to my question above about how many ineffective brown or white nodules with a T strain in them were found in plants grown in soil compared to effective nodules. In other words, T4 looks like it is getting into a nodule structure, which is quickly shut down by the plant. Is this a successful "trick" played by T4 to increase bacteria in the soil that helps increase long term numbers or is it a very rare example of nodulation gone wrong with little impact on soil numbers of the T strains.

-On a similar theme to the question above, what percentage of nodules were coinfecting with WSM419 and T4? I know there is, an albeit very poor photo claiming to show coinfection (Fig. 1E), but this may be a rare occurrence. Given the much higher competitive success of T4 (i.e. most nodules in a mixed inoculum are T4 not for example WSM419) I would not expect many coinfecting nodules. The key point is that the success of T4 does not appear to be dependent on coinfection as it forms more nodules than symbiotically effective rhizobia. However, here is a key point, if T4 is not released in significant numbers from nodules when it is the sole occupant, it may still gain an advantage by being present at a low frequency in effective nodules. Given that T4 appears to form many more nodules than WSM419 the coinfection route seems unlikely. Indeed, it seems to have a

developed a strategy of smash and run by getting into nodules very quickly (faster than successful effective strains). The plant quickly shuts these T4 nodules down but even if a much lower number of bacteria ultimately get out of these nodules (compared to WSM419) it may still result in a population increase of T4 in the soil.

-Line 45 “In this work, we report the characterization of the strain T4, an original endophytic bacterium.” The use of original is completely meaningless.

-I cannot see green Gfp labelled T4 in Fig 1B, resolution inadequate. The same applies to 1C. I can see green dots but whether they are in the same infection thread as purple WSM419 is beyond me. Figure 1D could be anything as I cannot see plant cells. What difference am I supposed to see between an infected cell and an uninfected cell in 1E?

-line 254. Give the cell densities in text even though it may be in methods? The cell density may be very important here and I think there needs to be a titration from 10^3 cfu per strain coinoculated up to higher number such as 10^6 . Higher numbers may be biologically meaningless. Checking the methods CFUs are not apparently given just OD600, where values of 0.01 and 0.1 are given for varying the effects of inoculation. These numbers are approximately 10^7 and 10^8 CFU, which are extremely high (Fig S12, although of course this depends on the exact size of these rhizobia and the inoculation volume (I just assume 1 ml), so my calculations may be wrong). The only time we use such high inoculations in my lab is when we want to force nodule co-occupancy.

-l 349 bacteroids are not released from infection threads, bacteria are released which then differentiate into bacteroids.

Reviewer #3 (Remarks to the Author):

This is a very comprehensive and detailed study of an *Ensifer adhaerens* strain, T4, that exhibits a wide range of age-dependent symbiotic/pathogenic phenotypes on a legume host, *Medicago truncatula*, ranging from (non-N-fixing) nodulation to plant death, with young plants being susceptible to damping-off symptoms leading to death, but older seedlings becoming resistant. T4 is also capable of being a very competitive nodulator, infecting *M. truncatula* and a wide range of related legumes in the IRLC via root hairs in a NF-dependent manner, and accordingly it possesses nod genes, but no nif genes (which fits with observations that the small nodules T4 forms are not symbiotically competent). It also appears that T4 is not alone within *E. adhaerens* in terms of this behavior, as at least one other strain, T173, has a dual nodulating/damping-off ability on *M. truncatula*. The data are discussed in terms of the evolution of nodulation from pathogens and the possibility that this has regressed in the case of T4 and T173, perhaps via “genetic erosion”. The manuscript is well written, the experiments were conducted in a highly competent manner, and the data are very novel. However, I would like to see more discussion of similar examples of non-symbiotic “cheater” rhizobia colonizing nodules. Indeed, I am particularly surprised that the study

of Zgad Zaj et al. (2015) has not been cited, as it describes in detail how a symbiotic rhizobium can enter and colonize *Lotus japonicus* nodules by “piggybacking” on the *Lotus* symbiont, *Mesorhizobium loti*, to avoid the host defense response.

Moreover, Cummings et al. (2009) described a naturally occurring fully functional N-fixing symbiont, strain IRBG74, now known to belong to the pathogenic species *Agrobacterium pusense*. IRBG74 appears to harbor *nod* and *nif* genes that were transferred from *Sesbania*-nodulating *Ensifer* spp.

Finally, although the micrographs are high quality, they lack the resolution needed to properly evaluate the ultrastructure of the nodules, particularly the bacteroids. Transmission electron microscopy (TEM) is needed for this, so please consider doing this.

Cummings et al. (2009) Nodulation of *Sesbania* species by Rhizobium (*Agrobacterium*) strain IRBG74 and other rhizobia. *Environmental Microbiology* 11: 2510-2525.

Zgad Zaj et al. (2015) A legume genetic framework controls infection of nodules by symbiotic and endophytic bacteria. *PLoS Genetics* DOI:10.1371/journal.pgen.1005280

Reviewer #4 (Remarks to the Author):

This is a very thorough manuscript presenting many interesting observations on a new strain of *Ensifer* that has negative effects on *Medicago* hosts in one of three ways depending on plant age: co-infect with fixing rhizobia, nodulate but not fix, and even kill juvenile plants. This strain also represents novel genomic information, in having 2 chromosomes and many plasmids. Unfortunately the current conceptual framing is insufficient and is not a good match to the data presented, which are generally descriptive of a new, albeit interesting, strain that essentially seems to be a plant pathogen with *nod* genes. The manuscript does not really pose or answer a biological question. The novelty could be more clear – how novel is this combination of traits in a bacterium? Which aspect is most novel, and why? Several unanswered conceptual questions remain in my mind: where did these *nod* genes come from? Is this a transient state due to recent HGT, or something more stable? Is this strain a pathogen compared to uninoculated plants? Again I do believe that these findings are of interest to the plant-microbe community, but without a more in-depth process-based analysis, the impact of this paper as currently assembled is somewhat limited.

ANSWERS TO REVIEWER COMMENTS

Dear reviewers,

Please find below, our point-by-point answers highlighted in blue. Please also find two versions of our revised manuscript 1) with modifications highlighted in yellow and 2) with all changes integrated.

Reviewer #1 (Remarks to the Author):

The manuscript by Magne et al. describe a novel strains of a root nodule endophytic bacterium able to colonize in a pseudorhizobial form root nodules and induce their formation on some host legumes. A detailed panel of biochemical, genomic and morphological analyses have been performed and clearly reported and commented. This discovery is giving momentum to the study of the evolution of symbiotic nitrogen fixation and deserve the merit to be published.

However, some minor points should be fixed and a more precise comment on the ecological context this strain could thrive should be given.

We would like to thank reviewer 1 for his/her positive feedback on the novelty and quality of our work. Please see below our answers to the different remarks raised by reviewer 1.

Specific points follow:

Line 71. Some check of English could be performed. E.g. a comma after “Despite that”.

Based on this remark, we did our best to polish the English language throughout the whole manuscript.

Line 113-114. Here you may better report previous findings and comments in the discussion section under the light of advantages of cheating and balancing selection for symbiosis vs. cheating. See for instance: <https://doi.org/10.1111/j.1574-6941.2007.00424.x>; <https://doi.org/10.1111/ele.13247> ; <https://doi.org/10.3389/fpls.2016.00835>.

We thank reviewer 1 for suggesting the following articles: Muresu et al., 2008; Checcucci et al., 2016 and Gano-Cohen et al., 2019. Initially we did not want to emphasize on the T4 endophytic behavior but to rather focus the discussion on the key pathogenic-nodulating aspects. However, based on reviewer 1's recommendation, we added an additional section related to the existence and the evolution of cheaters within the discussion.

Line 167. Is there one of the replicon a chromid? In other words, does it contains par system?

The second replicon can indeed be considered as a chromid. Based on the definition of a chromid from Harrison et al. in *Trend in Microbiology* 2010, the second replicon of the T4 meets the 3 criteria :

1) This second replicon possesses a rep-like system, with the gene b0001, b0002, b0003 being homologs with repA, repB and repC respectively. It contains also b1476, a parB-like gene and b1477 a putative repA copy.

2) The GC content of the second replicon is similar to the chromosome, i.e. with less than 1% deviation. Indeed, the GC content is 61.76% for the chromosome and 61.30% for the chromid, whereas the 5 plasmids range between 57.99% and 59.94%.

3) The second replicon contains few genes usually classified as housekeeping/essential genes. Indeed, we identified 12 out of the 1899 genes of the second replicon that can be classified as Minimal Gene Set based on the list of Gil et al. *Microbiol Mol Biol Rev* 2004.

Therefore, based on the characteristics of the T4's second replicon, we changed “chromosome 2” for “chromid” and “chromosome 1” for “chromosome” in the whole manuscript.

Figure 4 Q. The clade of *Ensifer sojae*-*E. medicae* should be named as *Sinorhizobium*. Please use the term *Sinorhizobium* along the text for members of this clade since *Ensifer* is not a valid name anymore for that clade (see your ref. n. 26 and paper in *IJSEM* <https://doi.org/10.1099/ijsem.0.005243>).

Based on the suggestion of reviewer 1 and considering both Fagorzi et al., 2020 [previous Ref 26] and Kuzmanovic et al., 2022 mentioned above by reviewer 1, we renamed the members of the symbiotic clade as “*Sinorhizobium*” and the members of the non-symbiotic clade as “*Ensifer*”. We applied these modifications in main text, figures, supplementary figures as well as sup. tables. Also, to better explain this important taxonomic point we updated the related section within the discussion section and cite the article from Kuzmanovic et al., 2022.

Line 548. Genome data availability must be reported as bioproject or similar on international databases (e.g. genbank or ENA).

T4 genome information are currently deposited on <https://mage.genoscope.cns.fr> with restricted access. However, as previously mentioned, the T4 genomic data will be publicly accessible on <https://mage.genoscope.cns.fr> following publication acceptance. Furthermore, once published, we will ask <https://mage.genoscope.cns.fr> to deposit the T4 genomic data on NCBI under the bioproject ID PRJNA1066792. These information have been added in the data availability section of the manuscript.

Line 549. Since the importance of such a strain a deposit on a publicly accessible strain repository (e.g. DSMZ) must be done.

We made the *E. adhaerens* T4 strain available at the CIRM-CFBP French Collection for Plant Associated Bacteria (<https://cirm-cfbp.fr/>) under the accession number CFBP 9181. We added this information in the material availability section.

Fig. S16. A more readable representation could be better, why not to use a heatmap or similar?

As biolog charts provide kinetic information that are lost when the data are presented as a heat map, we preferred to keep biolog charts (Supplementary Figures 16), and according to the reviewer 1 recommendation, we added two additional supplementary figures (Supplementary Figures 17 and 18) representing the same PM01 and PM02 carbon source-based biolog data as heatmaps.

Table S2. Please do not copy and past an excel datasheet without any legend and with some columns names partially masked! Table S6. This table is mostly unreadable. The same for Table S7.

For the initial submission of our manuscript to Nature Communications, we indeed provided a merged pdf containing all the data, including supplementary excel tables as pictures. We do recognize that this was not optimal and we do apologize for this. In this new version of the manuscript we provided all supplementary tables as .xlsx files.

Reviewer #2 (Remarks to the Author):

This is a very thorough and interesting piece of work that suggests that T4 is a pathogenic strain of Ensifer that has nod but not nif genes. It would appear the acquisition of nod genes allows T4 to form aborted infections on Medicago spp. I have specific questions below but the thrust of my comments is about the effects on population size of T4 in the possession of nod genes, but in the absence of nif genes and N₂ fixation and mutualistic cooperation with the plant. My guess is that T4 is quickly shut down by the plant in nodules and therefore is released from nodules in much smaller numbers (if at all) than a symbiotically effective strain. Just as we see for poorly effective strains of Rhizobia (that possess nod and nif genes), sanctioning by the plant means these cheating strategies may work up to a point but such strains are kept to a low frequency. However, this of course is my speculation because these experiments are not shown in the work so far.

We would like to thank reviewer 2 for his/her positive feedback on our work. Based on reviewer 2 comments, we performed additional experiments aiming to re-isolate T4 and WSM419 from surface-sterilized nodules. These experiments were performed, using *M. truncatula* A17, which triggers Terminal Differentiation of Bacteroids (TDB, see Mergaert et al., 2006; DOI: 10.1073/pnas.0600912103). This means that effective rhizobia, once internalized and differentiated, cannot return to a free-living state. In this paper, authors mentioned that it is particularly challenging to isolate rhizobia from nodules. Coming back to our experiment, we succeeded in re-isolating viable T4 from white and brown T4-derived nodules at 21-dpi to an extent that was similar to higher than for WSM419 (please see table below).

		big pink WSM419	small white T4 nodule	small brown T4 nodule
Biological repeat 1	Control (8th wash)	0/4	0/5	0/5
	Nodule pool (pool of 5 nodules)	0/4	0/5	3/5
Biological repeat 2	Control (8th wash)	0/6	0/9	0/8
	Nodule pool (pool of 5 nodules)	2/6	3/9	4/8

First of all, please notice that in these experiments, all controls were free of any viable bacteria meaning that our surface sterilization procedure is accurate and that reisolated bacteria correspond to true nodule endophytes. Because of the TDB process, the rare re-isolated WSM419 would likely come from not yet differentiated WSM419 residing within infection threads. Regarding the re-isolated T4, we do not know whether they represent offspring of bacteria internalized within the host cells or apoplasmic bacteria residing within the infection threads. Nevertheless, we observed that it was much easier to re-isolate T4 than WSM419 from nodules and we suspect that the non-differentiation of the T4 likely contributes to its capacity to return to the free-living state.

In addition, it is worth to note that the ecological success of T4 does not only rely on its ability to resume growth after nodulation, but also on its ability to i) trigger nodules (we showed that T4 is more efficient for this than 20 other tested rhizobial strains (Fig. 2), ii) to develop as a pathogen on young seedlings (please see the new panel we added showing the dynamic of T4 proliferation on whole plant under pathogenic behavior (Fig. 4J) as well as iii) on its ability to multiply and survive outside the host in the soil. We now discussed these aspects within the discussion section.

-It is not clear where T4 and indeed T1 and T3 were isolated from. Were they isolated as small white or brown nodules or from coinfecting nodules? I suspect individual small white/brown nodules as these would be easy to find but I am not sure from the paper.

The *E. adhaerens* T4, T1 and T2 as well as the other *Sinorhizobium medicae* and *Microbacterium* sp. strains described in the Supplementary Table 1 have been isolated from pools of surface-sterilized nodules which contained nitrogen-fixing nodules as well as likely non-fixing white and brown nodules. We integrated this information within the material and method section.

This leads directly to my next question which is what percentage of nodules in the original soil inoculum had these ineffective T4 strains? This is actually a crucial ecological question as such parasitic/pathogenic strains would be very harmful to *Medicago* spp. if present at high levels in soil.

We are unfortunately not able to answer this point since isolations of nodule endophytes were performed from pools of nodules. In addition, we did not perform a precise nodule phenotyping at the initial trapping step as we did not expect at this time to isolate such bacteria. We do agree that the existence of such parasitic strains in French and Canadian soil raises some sanitary questions and it would definitely be worth evaluating both the presence and the quantity of comparable strains in different geographic areas. Since we believe that this point is relevant, we added a comment on this within the discussion.

-A question of major importance is how many T4 bacteria regrow from the ineffective nodules over a time course and crucially how does this compare to an effective strain? The time course also needs to be run for a reasonable long time such as 6-8 weeks to replicate senescence. The reason for this question is that for this to be an evolutionary successful strategy T4 must get released from nodules and be able to regrow. I would expect there to be orders of magnitude more rhizobia released from effective nodules (e.g. WSM419 infected) than T4 infected nodules. This also relates to my question above about how many ineffective brown or white nodules with a T strain in them were found in plants grown in soil compared to effective nodules. In other words, T4 looks like it is getting into a nodule structure, which is quickly shut down by the plant. Is this a successful "trick" played by T4 to increase bacteria in the soil that helps increase long term numbers or is it a very rare example of nodulation gone wrong with little impact on soil numbers of the T strains.

We partially addressed this question above, right after the general comment of reviewer 2. So, as argued before, effective rhizobia undergo TDB within *Medicago* nodules and are thus unable to return to the free-living state. We also wanted to point out the fact that it is technically challenging to sterilize T4 nodules because of their small size and their high sensibility/vulnerability to surface sterilization treatment (EtOH 70% for 1 min followed by NaOCl 6% for 3 min). In addition, as previously mentioned above, it seems that already at early stages of development (21 dpi), the T4 strain shows a better capacity to exit and survive from nodules. Once more, it is worth to point out that even if the T4 strain is indeed able to form nodules in a more effective way than other known efficient and inefficient symbionts, to multiply inside and to exit those nodules, the pathogenic lifestyle of the T4 strain on young seedlings would largely contribute to the ecological success of this strain.

-On a similar theme to the question above, what percentage of nodules were coinfecting with WSM419 and T4? I know there is, an albeit very poor photo claiming to show coinfection (Fig. 1E), but this may be a rare occurrence. Given the much higher competitive success of T4 (i.e. most nodules in a mixed inoculum are T4 not for example WSM419) I would not expect many coinfecting nodules. The key point is that the success of T4 does not appear to be dependent on coinfection as it forms more nodules than symbiotically effective rhizobia. However, here is a key point, if T4 is not released in significant numbers from nodules when it is the sole occupant, it may still gain an advantage by being present at a low frequency in effective nodules. Given that T4 appears to form many more nodules than WSM419 the coinfection route seems unlikely. Indeed, it seems to have developed a strategy of smash and run by getting into nodules very quickly (faster than successful effective strains). The plant quickly shuts these T4 nodules down but even if a much lower number of bacteria ultimately get out of these nodules (compared to WSM419) it may still result in a population increase of T4 in the soil.

We did not precisely quantify the percentage of nodules infected by both WSM419 and T4. However, based on laser scanning confocal microscopy experiments, we only observed co-infection events within big pink nitrogen fixing nodules derived from WSM419, also, these big pink WSM419 nodules were rather rare under mixed-inoculation with T4 (this is clearly shown by competition assays in Fig. 2 and Supplementary Fig. 6). Given the higher success of the T4 strain versus other symbionts for nodule formation, this makes these co-infection events quite rare. We added a sentence, within the result section dealing with co-infection events, in order to clarify this point. Also, our data indicate that T4 can return free living post nodule infection (please see the different

answers above on this subject), however we do not know the fate of T4 bacteria in the case of these rare co-infected nodules.

-Line 45 "In this work, we report the characterization of the strain T4, an original endophytic bacterium." The use of original is completely meaningless.

For the revised version of the manuscript, the whole abstract has been rewritten to fit with the journal guideline (150 words max.) and the word "original" has been removed. In the manuscript, we also replaced the word "original" for "atypical" when necessary.

-I cannot see green Gfp labelled T4 in Fig 1B, resolution inadequate. The same applies to 1C. I can see green dots but whether they are in the same infection thread as purple WSM419 is beyond me. Figure 1D could be anything as I cannot see plant cells. What difference am I supposed to see between an infected cell and an uninfected cell in 1E?

We initially submitted the first version of our manuscript as a single .PDF document and indeed we realized that the resolution of pictures was lower than expected. We have now submitted a new version of the manuscript with pictures having higher resolution and we believe that it is now possible to identify the objects. Please note that for Fig. 1D, we slightly increased the luminosity of the picture in order to better see the plant cell background and near infected cells. Also, for Fig. 1E, we indeed realized that infected cell (ic) and uninfected cell (uc) were misplaced in the picture, please accept our apologies for this issue. We corrected this mistake.

-line 254. Give the cell densities in text even though it may be in methods? The cell density may be very important here and I think there needs to be a titration from 10^3 cfu per strain coinoculated up to higher number such as 10^6 . Higher numbers may be biologically meaningless. Checking the methods CFUs are not apparently given just OD600, where values of 0.01 and 0.1 are given for varying the effects of inoculation. These numbers are approximately 10^7 and 10^8 CFU, which are extremely high (Fig S12, although of course this depends on the exact size of these rhizobia and the inoculation volume (I just assume 1 ml), so my calculations may be wrong). The only time we use such high inoculations in my lab is when we want to force nodule co-occupancy.

-To answer this point, we indeed expressed bacterial densities using OD. We did additional experiments and determined the correspondence between OD and cfu for WSM419 and T4 strains. We evaluated using standardized plating (easySpiral) that for both WSM419 and T4, OD_{600nm} : 0.1 corresponds to $1,32-1,56 \cdot 10^8$ cfu/mL and to $2,14-2,20 \cdot 10^8$ cfu/mL for exponential and stationary phases, respectively. This order of magnitude was expected for rhizobia. In order to not overload the text with methodological information, we preferred to provide these information within the methods.

-Regarding the point about the titration of bacterial suspension used for co-inoculation experiments, please note that in supplementary data Fig. 6G, we tested OD_{600nm} of 0.1, 0.01, 0.001 and 0.0001 for the T4 strain against WSM419 at OD_{600nm} of 0.1 and obtained similar results.

-Reviewer 2 also raised the point that titration may also vary according to the cell size of bacterial strains. To solve this question, we performed additional experiments and determined the dry weight of both WSM419 and T4 per mL at an OD_{600nm} of 0.1. As a result, we did not observe a significant difference between WSM419 and T4. Please see the corresponding results below.

-l 349 bacteroids are not released from infection threads, bacteria are released which then differentiate into bacteroids.

We have now modified the corresponding sentence by the following : “Remarkably, even at 21 dpi, while most T4 bacteroids were dead, some alive T4 bacteroids were detected, likely representing freshly released bacteria from infection threads”.

Reviewer #3 (Remarks to the Author):

This is a very comprehensive and detailed study of an *Ensifer adhaerens* strain, T4, that exhibits a wide range of age-dependent symbiotic/pathogenic phenotypes on a legume host, *Medicago truncatula*, ranging from (non-N-fixing) nodulation to plant death, with young plants being susceptible to damping-off symptoms leading to death, but older seedlings becoming resistant. T4 is also capable of being a very competitive nodulator, infecting *M. truncatula* and a wide range of related legumes in the IRLC via root hairs in a NF-dependent manner, and accordingly it possesses nod genes, but no nif genes (which fits with observations that the small nodules T4 forms are not symbiotically competent). It also appears that T4 is not alone within *E. adhaerens* in terms of this behavior, as at least one other strain, T173, has a dual nodulating/damping-off ability on *M. truncatula*. The data are discussed in terms of the evolution of nodulation from pathogens and the possibility that this has regressed in the case of T4 and T173, perhaps via “genetic erosion”. The manuscript is well written, the experiments were conducted in a highly competent manner, and the data are very novel. However, I would like to see more discussion of similar examples of non-symbiotic “cheater” rhizobia colonizing nodules. Indeed, I am particularly surprised that the study of Zgadzaj et al. (2015) has not been cited, as it describes in detail how a symbiotic rhizobium can enter and colonize *Lotus japonicus* nodules by “piggybacking” on the *Lotus* symbiont, *Mesorhizobium loti*, to avoid the host defense response. Moreover, Cummings et al. (2009) described a naturally occurring fully functional N-fixing symbiont, strain IRBG74, now known to belong to the pathogenic species *Agrobacterium pusense*. IRBG74 appears to harbor nod and nif genes that were transferred from *Sesbania*-nodulating *Ensifer* spp.

We thank reviewer 3 for suggesting us to cite the work of Zgadzaj et al. 2015. We now cited this article within the discussion. Since nodule co-infection events are rare in our system, we do not want to speculate about the development of defense reactions in co-infected nodules.

We also thank reviewer 3 for pinpointing the relevant case of IRBG74. We therefore introduced the IRBG74 strain in the discussion of our manuscript and integrated this strain within our phylogenetic analysis (Fig. 1) together with the C58 pathogenic sister strain. Now one can read in the discussion: “In addition, phylogenetic proximity was observed between pathogenic and symbiotic bacteria with sometimes blurry boundaries. For instance, the non-pathogenic strain IRBG74 is genomically very close to pathogenic *Agrobacteria* but it nodulates *sesbania* (Cummings et al. 2009)”.

Finally, although the micrographs are high quality, they lack the resolution needed to properly evaluate the ultrastructure of the nodules, particularly the bacteroids. Transmission electron microscopy (TEM) is needed for this, so please consider doing this.

For the initial submission, the resolution of pictures unfortunately became sub-optimal at the .pdf transformation step. For this resubmission, we now provided all pictures with a higher resolution on which bacteroids are now more visible. Also, we think that the evaluation of nodule ultrastructure using TEM will neither support or invalidate the conclusions of our manuscript and might therefore be beyond the scope of our story.

Cummings et al. (2009) Nodulation of *Sesbania* species by Rhizobium (*Agrobacterium*) strain IRBG74 and other rhizobia. *Environmental Microbiology* 11: 2510-2525.

Zgadzaj et al. (2015) A legume genetic framework controls infection of nodules by symbiotic and endophytic bacteria. *PLoS Genetics* DOI:10.1371/journal.pgen.1005280

Reviewer #4 (Remarks to the Author):

This is a very thorough manuscript presenting many interesting observations on a new strain of *Ensifer* that has negative effects on *Medicago* hosts in one of three ways depending on plant age: co-infect with fixing rhizobia, nodulate but not fix, and even kill juvenile plants. This strain also represents novel genomic information, in having 2 chromosomes and many plasmids. Unfortunately the current conceptual framing is insufficient and is not a good match to the data presented, which are generally descriptive of a new, albeit interesting, strain that essentially seems to be a plant pathogen with nod genes. The manuscript does not really pose or answer a biological question. The novelty could be more clear – how novel is this combination of traits in a bacterium? Which aspect is most novel, and why? Several unanswered conceptual questions remain in my mind: where did these nod genes come from? Is this a transient state due to recent HGT, or something more stable? Is this strain a pathogen compared to uninoculated plants? Again I do believe that these findings are of interest to the plant-microbe community, but without a more in-depth process-based analysis, the impact of this paper as currently assembled is somewhat limited.

We, first of all, would like to thank reviewer 4 for highlighting the scientific interest of our findings.

-The novelty could be more clear – how novel is this combination of traits in a bacterium?

-Which aspect is most novel, and why?

In the initial version of our abstract one can read: “bacterial strains having maintained both pathogenic and nodulation features on a single host have not been described yet.” and “We found that T173 and T4 share the common dual ability to trigger nodulation and plant death in various species.” We expected that these sentences clearly show the novelty of our study. To better emphasize on the novelty of our work, the discussion now includes the following sentence : “In addition, phylogenetic proximity was observed between pathogenic and symbiotic bacteria with sometimes blurry boundaries. For instance, the non-pathogenic strain IRBG74 is genomically very close to pathogenic *Agrobacteria* but it nodulates *sesbania* (Cummings et al. 2009).”, just before the following initial statement: “We isolated and characterized an atypical bacterial strain, namely *Ensifer adhaerens* T4 (T4), that naturally displays the ability to trigger severe disease and to induce nodules on a single legume host species depending on the plant developmental stage.”

As answered to reviewer 2, we also included new results to give more emphasis on the pathogenic behavior of *Ensifer* T4 strain which is the main novelty of our study (please see, new panel in figure 4).

-where did these nod genes come from?

In the previous version of our manuscript, within the result section, one can read : “Blast searches against the NCBI nucleotide database indicated that genes involved in the production, decoration and secretion of T4 NFs were almost identical to those of T173 and WSM419 (Supplementary Fig. 3). Phylogenetic reconstruction based on *nodABCIIJ* genes indicated that T4, T173 and WSM419 genes form a cluster distinct from that of *Sinorhizobium meliloti* strains (Supplementary Fig. 3). The proximity between WSM419 and T4 *nod* genes was further confirmed by comparing the structure of WSM419 and T4 NFs (Supplementary Fig. 4).”. Furthermore, in the discussion section we initially stated that: “The organization of T4 and T173 genomes, especially for replicons carrying *nod* genes, is substantially different suggesting old and divergent or independent horizontal acquisition.” and we also added, right after this sentence, that: “Based on sequence analysis (Supplementary Fig. 3), it seems reasonable to speculate that the *E. adhaerens* T4 and T173 nodulation genes were acquired from either one or two distinct *S. medicae* strains.”

- Is this a transient state due to recent HGT, or something more stable?

This is a puzzling question, in the previous version one could read in the discussion section: “The organization of T4 and T173 genomes, especially for replicons carrying nod genes, is substantially different suggesting old and divergent or independent horizontal acquisition.” We proposed to modify this sentence by the following one: “The organization of T4 and T173 genomes, especially for replicons carrying nod genes, is substantially different

suggesting either old and divergent or independent and more recent horizontal acquisitions.” In addition, the question of the plasmid instability was mentioned in the previous version of the manuscript and we now provide additional arguments in the discussion : “Furthermore, these two plasmids display a high proportion of repeated regions (11% and 27% for T173 and T4, respectively) which is a typical feature of instability and suggests an ongoing genetic erosion of those replicons. Nevertheless, until now T4 and T173 symbiotic plasmids have been maintained. The capacity of these atypical *E. adhaerens* strains to form and colonize nodules at high frequency without fixing nitrogen might represent an evolutionary asset allowing the diversification of *E. adhaerens* ecological niches.”

- Is this strain a pathogen compared to uninoculated plants?

Yes it is. For instance, the data presented in the Fig. 4A have been obtained through the comparison of T4-inoculated plants with mock- inoculated plants.

REVIEWER COMMENTS

Reviewer #1 (Remarks to the Author):

The manuscript by Magne et al. has been strongly improved by the authors. Availability to strains and data obtained and a more complete discussion of results with respect to the current knowledge on this bacterial group are now present.

Reviewer #2 (Remarks to the Author):

Reviewer general comment and question beginning “A question of major importance is how many T4 bacteria regrow from the ineffective nodules over a time course and crucially how does this compare to an effective strain?”

The authors replies on reisolation of strains from nodules are a complete failure to answer the questions set out. The reisolations cannot be performed at 21 d as nodules do not naturally senesce at this very early time point. At such an early time point it is normal to be able to isolate more bacteria from ineffective nodules (e.g. T4) rather than effective nodules (e.g. WSM419) because they do not differentiate properly and tend to remain in infection threads. The only reason for using a time point of 21 d is that it will give you the answer you want (i.e. bacteria that do not differentiate and are ineffective will have more viable bacteria at an early time point) rather than representing how many bacteria will be released at a time point when nodules would naturally senesce. At later time points the number of bacteria in ineffective nodules drop rapidly because the nodules are sanctioned by the plant. The plant cuts off the nutrient supply to ineffective nodules after 12-16 d and the numbers of bacteria that are still viable at later timepoints of approximately 45-65 d (exact time depends on the species of plant) drop rapidly. Effective nodules are not sanctioned so the numbers of bacteria released at later time points (i.e. the sort of ages when nodules might naturally break down and release bacteria) stay high. This is why I so explicitly stated the number of bacteria isolated from each nodule must be determined at late time points (6-8 weeks in my original comments). They should also be as colony forming units per nodule, which are typically 10^6 - 10^8 cfu/nodule.

There is a strange table giving ratios numbers of pooled isolates such as 2/6 or 3/9 which are number numbers of bacteria reisolated from nodules. I can only imagine this means that 2 out of 6 nodules yielded reisolated bacteria rather than giving counts of bacteria i.e. bacterial cfu per nodule. There is a very serious problem if you cannot isolate bacteria from 100% of effective nodules. It is stated that it is difficult to isolate bacteria from nodules and this is just not true. While it is true that you cannot isolated viable bacteroids from nodules of differentiated bacteroids from

the IRLC clade, there are very large numbers of bacteria present in infection threads that can be easily isolated. If the authors are struggling to re-isolate wildtype bacteria (e.g. WSM419) from nodules of *M. truncatula* A17 then use another legume such as *Medicago sativa*, which is a much more robust plant than *M. truncatula* but recognises the same Nod factors.

Furthermore, it is stated in the first page of the reply to my questions “In addition, it is worth to note that the ecological success of T4 does not only rely on its ability to resume growth after nodulation, but also on its ability to i) trigger nodules (we showed that T4 is more efficient for this than 20 other tested rhizobial strains (Fig. 2), ii)”. The whole point of my questions were that the ultimate competitive success of T4 depends not just on its ability to get into nodules (which I accept is clearly very good) but it depends on its ability to get out of nodules after sanctioning by the plant, which is clearly very, very, very severe in the case of T4 (that is why nodules containing T4 are either very small or white and brown).

(Reviewer question) This leads directly to my next question which is what percentage of nodules in the original soil inoculum had these ineffective T4 strains? This is actually a crucial ecological question as such parasitic/pathogenic strains would be very harmful to *Medicago* spp. if present at high levels in soil.

The authors answer is that careful nodule phenotyping was not done originally as the nodules were pooled to obtain isolates. I accept that this is not uncommon but what is the reason for not going back and doing careful nodule phenotyping now? There is a fundamental problem with this work, which comes back to the question of ecological significance. Yes, I accept that T4 has developed an interesting strategy to cheat the plant based on possessing nod genes but not providing nitrogen to the plant (i.e. no *nif*) but at the moment this just looks like a very rare curiosity. There is no evidence in this work that this strategy is common. The authors have ducked the nodule recovery experiments (see above) and are ducking the vital questions of how frequent is this in real soils? Legumes sanction ineffective nodules to prevent run away infection by cheating strains and I see no evidence that T4 is able to escape nodule sanctioning. All the evidence so far in this paper is that T4 will never be at very high frequency in soil as it has a strategy to get into nodules but will not come out of senescent nodules at high frequency. It is a great strategy to occupy a rare ecological niche where it is present at low but stable frequency if legumes are growing in the soil.

Reviewer #4 (Remarks to the Author):

The revision has sufficiently addressed the many comments of four diverse reviewer perspectives.

THE AUTHORS: Please find below our answers to reviewer comments highlighted in magenta. Please also find two versions of our revised manuscript 1) with modifications highlighted in yellow and 2) with all changes integrated.

Reviewer #1 (Remarks to the Author):

The manuscript by Magne et al. has been strongly improved by the authors. Availability to strains and data obtained and a more complete discussion of results with respect to the current knowledge on this bacterial group are now present.

THE AUTHORS: We thank reviewer #1 for her/his previous comments, advices and suggestions which allowed us a significant improvement of our manuscript.

Reviewer #2 (Remarks to the Author):

Reviewer general comment and question beginning “A question of major importance is how many T4 bacteria regrow from the ineffective nodules over a time course and crucially how does this compare to an effective strain?”

The authors replies on reisolation of strains from nodules are a complete failure to answer the questions set out. The reisolations cannot be performed at 21 d as nodules do not naturally senesce at this very early time point. At such an early time point it is normal to be able to isolate more bacteria from ineffective nodules (e.g. T4) rather than effective nodules (e.g. WSM419) because they do not differentiate properly and tend to remain in infection threads. The only reason for using a time point of 21 d is that it will give you the answer you want (i.e. bacteria that do not differentiate and are ineffective will have more viable bacteria at an early time point) rather than representing how many bacteria will be released at a time point when nodules would naturally senesce. At later time points the number of bacteria in ineffective nodules drop rapidly because the nodules are sanctioned by the plant. The plant cuts off the nutrient supply to ineffective nodules after 12-16 d and the numbers of bacteria that are still viable at later timepoints of approximately 45-65 d (exact time depends on the species of plant) drop rapidly. Effective nodules are not sanctioned so the numbers of bacteria released at later time points (i.e. the sort of ages when nodules might naturally break down and release bacteria) stay high. This is why I so explicitly stated the number of bacteria isolated from each nodule must be determined at late time points (6-8 weeks in my original comments). They should also be as colony forming units per nodule, which are typically 10^6 - 10^8 cfu/nodule. There is a strange table giving ratios numbers of pooled isolates such as 2/6 or 3/9 which are number numbers of bacteria reisolated from nodules. I can only imagine this means that 2 out of 6 nodules yielded reisolated bacteria rather than giving counts of bacteria i.e. bacterial cfu per nodule. There is a very serious problem if you cannot isolate bacteria from 100% of effective nodules. It is stated

that it is difficult to isolate bacteria from nodules and this is just not true. While it is true that you cannot isolated viable bacteroids from nodules of differentiated bacteroids from the IRLC clade, there are very large numbers of bacteria present in infection threads that can be easily isolated. If the authors are struggling to reisolate wildtype bacteria (e.g. WSM419) from nodules of *M. truncatula* A17 then use another legume such as *Medicago sativa*, which is a much more robust plant than *M. truncatula* but recognises the same Nod factors. Furthermore, it is stated in the first page of the reply to my questions “In addition, it is worth to note that the ecological success of T4 does not only rely on its ability to resume growth after nodulation, but also on its ability to i) trigger nodules (we showed that T4 is more efficient for this than 20 other tested rhizobial strains (Fig. 2), ii)”. The whole point of my questions were that the ultimate competitive success of T4 depends not just on its ability to get into nodules (which I accept is clearly very good) but it depends on its ability to get out of nodules after sanctioning by the plant, which is clearly very, very, very severe in the case of T4 (that is why nodules containing T4 are either very small or white and brown).

THE AUTHORS: We thank the reviewer #2 for her/his interest on the ecological success of the T4 and for raising the following point: “how many T4 bacteria regrow from ineffective nodules over a time course and how does this compare to an effective strain?”. This is an interesting question, however, we regret that reviewer #2 did not quote our whole answer and neglected the second and third points of our answer (please see the sentence below): “In addition, it is worth to note that the ecological success of T4 does not only rely on its ability to resume growth after nodulation, but also on its ability to i) trigger nodules (we showed that T4 is more efficient for this than 20 other tested rhizobial strains (Fig. 2), ii) to develop as a pathogen on young seedlings (please see the new panel we added showing the dynamic of T4 proliferation on whole plant under pathogenic behavior (Fig. 4J) as well as iii) on its ability to multiply and survive outside the host in the soil”.

This being said, we apologize for having initially not paid enough attention to this point during the first round of reviewing. We understand that it deserves more attention and therefore we performed additional experiments with different setups in order to provide appropriate and convincing elements to answer this question, namely, how many T4 and effective bacteria are released from a single nodule at a late time point when nodules senesce?

We were quite struggling with how to address this point since we were facing two major challenges:

- 1) The first issue we had, at such late time points suggested by reviewer #2 (6-8 weeks post inoculation), was the following: the T4-inoculated plants continuously starved for nitrogen and ultimately died. As a demonstration, the pictures below show plants at 5 weeks post inoculation. At this time, T4-inoculated plants are suffering from a severe nitrogen starvation (T4 lacks nitrogen fixing genes and is therefore strictly fix minus) and start to die. In our *in vitro* system, at 5 weeks post inoculation, we reached a limit for analyzing the T4-inoculated plants in comparison

with efficient bacteria undergoing natural and late senescence. Please see in the figure below that T4-inoculated plants are very different from those inoculated with the efficient symbiotic bacterium WSM419.

T4- and WSM-inoculated plants 5 weeks post inoculation. T4-inoculated plants are suffering from a severe nitrogen starvation and start to die.

2) The second challenge we faced was that the sterilization process is very difficult to control, especially for small and often necrotic nodules induced by T4. Indeed, in similar samples and depending on how the T4 nodule structure is altered by the necrosis, the sterilizing compound can penetrate the interior of the nodule and be too strong. Inversely, a faster or softer procedure can be too weak and generate false results leading to wrong interpretations.

Considering these two challenges, we performed different approaches to solve this point. This includes *in vitro* and non-*in vitro* experiments. We also analyzed senescent nodules following natural and dark-induced senescence. As *in vitro* nodules are more sensitive to sterilization treatment, we used a mild surface sterilization protocol (70% ethanol for 15 seconds) for plants grown in these conditions.

1) We first counted cfu from single T4 and WSM419 nodules at 3 and 5 weeks post inoculation *in vitro*. 5 weeks post inoculation, WSM419 nodules presented a senescence phenotype (greenish, please see pictures below). Here, nodules were surface-sterilized by 70% ethanol for 15 s.

WSM- and T4- nodules 5 weeks post inoculation. 5 weeks post inoculation WSM419 nodules showing a senescence phenotype (greenish coloration).

2) We also counted cfu from single T4 and WSM419 nodules at 3 weeks post inoculation *in vitro* but on which senescence was induced by a darkness treatment for 3 days. After this treatment, WSM419 nodules were senescent (greenish). Here again, nodules were surface-sterilized by 70% ethanol for 15 s.

3) We also counted cfu from single T4 and WSM419 nodules at 3 weeks post inoculation of plants grown in a sand and perlite mixture on which senescence was induced by a darkness treatment for 3 days. After this treatment, WSM419 nodules were senescent (greenish). Here, nodules were surface-sterilized by 2 min water/surfactant and 2 min 2.6% sodium hypochlorite.

As a result, our experiments indicate that both WSM419 and T4 strains can regrow from nodules following natural or induced senescence. It can be noticed that such a result was also obtained with the harsher sterilization treatment consisting of 2 min water/surfactant and 2 min 2.6 % sodium hypochlorite, with plants grown either *in vitro* or in a sand and perlite mixture. Also, as expected by reviewer #2, up to 10^6 - 10^8 cfu/nodule were recovered from both WSM419 and T4 senescent nodules under *in vitro* condition. A reduced bacterial titre was obtained for both WSM419 and T4 nodules grown in peat and in the sand and perlite mixture.

We included these additional information in the manuscript discussion within the section dedicated to T4 ecological success and integrated a related supplementary figure to support our statements. Please see changes we made highlighted in yellow in the manuscript tracking changes as well as the Supplementary Fig. 21.

(Reviewer question) This leads directly to my next question which is what percentage of nodules in the original soil inoculum had these ineffective T4 strains? This is actually a crucial ecological question as such parasitic/pathogenic strains would be very harmful to *Medicago* spp. if present at high levels in soil. The authors answer is that careful nodule phenotyping was not done

originally as the nodules were pooled to obtain isolates. I accept that this is not uncommon but what is the reason for not going back and doing careful nodule phenotyping now? There is a fundamental problem with this work, which comes back to the question of ecological significance. Yes, I accept that T4 has developed an interesting strategy to cheat the plant based on possessing nod genes but not providing nitrogen to the plant (i.e. no nif) but at the moment this just looks like a very rare curiosity. There is no evidence in this work that this strategy is common. The authors have ducked the nodule recovery experiments (see above) and are ducking the vital questions of how frequent is this in real soils? Legumes sanction ineffective nodules to prevent run away infection by cheating strains and I see no evidence that T4 is able to escape nodule sanctioning. All the evidence so far in this paper is that T4 will never be at very high frequency in soil as it has a strategy to get into nodules but will not come out of senescent nodules at high frequency. It is a great strategy to occupy a rare ecological niche where it is present at low but stable frequency if legumes are growing in the soil.

THE AUTHORS: Concerning this second point raised by reviewer #2, which is, to go back to the original soil, to repeat inoculation with the soil and to determine the percentage of ineffective T4 nodules on the plant root system. It is unfortunately now eight years (September 2016) since the original soil was sampled. The original soil is no longer in our possession. Nevertheless, in order to answer reviewer #2's request, we collected new soil (beginning of April 2024) at the same site and exactly repeated the original bacterial trapping experiment as described within the methods of our manuscript. For this experiment 30 plants were inoculated with soil. 35 days post inoculation, surprisingly, we did not get neither inefficient nor efficient nodules on any of the plants. Please note that our experiment included a positive control which consisted of 15 additional WSM419-inoculated plants that were all correctly nodulated.

It is possible to explain this unexpected discrepancy by different reasons. First, the sampling season between the original sampling of September 2016 and the present sampling of April 2024 is different. This might drastically impact the structure of microbial populations. Second, microbial populations might also change from a year to another and likely evolved in 8 years. We are regretful for not being able to reisolate T4 from the same location a second time.

In regard with the following statement of reviewer #2 : *"Yes, I accept that T4 has developed an interesting strategy to cheat the plant based on possessing nod genes but not providing nitrogen to the plant (i.e. no nif) but at the moment this just looks like a very rare curiosity. There is no evidence in this work that this strategy is common."*. We think that this comment is inappropriate since in this manuscript we reported a similar behavior for the closely related strain T173. It worths to note that the T173 strain has been isolated from another continent, from another plant species and that T173 is genomically distinct from T4 (please see Fig 5; Supp. Fig. 20 and Venn diagram on Supp. Fig 15).

Reviewer #4 (Remarks to the Author):

The revision has sufficiently addressed the many comments of four diverse reviewer perspectives.

THE AUTHORS: We thank the reviewer #4 for her/his previous comments, advices and suggestions as well as for having raised important points. This allowed to significantly improve our manuscript.